# HDR-NSFF: High Dynamic Range Neural Scene Flow Fields

**Shin Dong-Yeon**[1]    **Kim Jun-Seong**[2]    **Kwon Byung-Ki**[3]    **Tae-Hyun Oh**[4]

[1]School of Electrical Engineering, KAIST    [2]Dept. of Electrical Engineering, POSTECH
[3]Grad. School of Artificial Intelligence, POSTECH    [4]School of Computing, KAIST

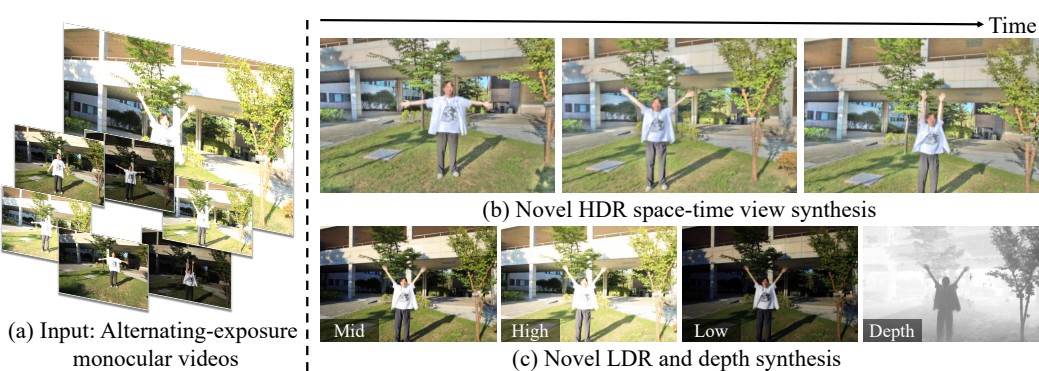

Figure 1: **High Dynamic Range Neural Scene Flow Fields (HDR-NSFF)** reconstruct dynamic HDR radiance field from (a) alternating-exposure monocular videos. Our method enables the rendering of (b) HDR novel views across both spatial and temporal domains. Additionally, we can generate (c) novel LDR views along with their corresponding depth maps.

## Abstract

Radiance of real-world scenes typically spans a much wider dynamic range than what standard cameras can capture. While conventional HDR methods merge alternating-exposure frames, these approaches are inherently constrained to 2D pixel-level alignment, often leading to ghosting artifacts and temporal inconsistency in dynamic scenes. To address these limitations, we present HDR-NSFF, a paradigm shift from 2D-based merging to 4D spatio-temporal modeling. Our framework reconstructs dynamic HDR radiance fields from alternating-exposure monocular videos by representing the scene as a continuous function of space and time, and is compatible with both neural radiance field and 4D Gaussian Splatting (4DGS) based dynamic representations. This unified end-to-end pipeline explicitly models HDR radiance, 3D scene flow, geometry, and tone-mapping, ensuring physical plausibility and global coherence. We further enhance robustness by (i) extending semantic-based optical flow with DINO features to achieve exposure-invariant motion estimation, and (ii) incorporating a generative prior as a regularizer to compensate for limited observation in monocular captures and saturation-induced information loss. To evaluate HDR space-time view synthesis, we present the first real-world HDR-GoPro dataset specifically designed for dynamic HDR scenes. Experiments demonstrate that HDR-NSFF recovers fine radiance details and coherent dynamics even under challenging exposure variations, thereby achieving state-of-the-art performance in novel space-time view synthesis. Project page: https://shin-dong-yeon.github.io/HDR-NSFF/

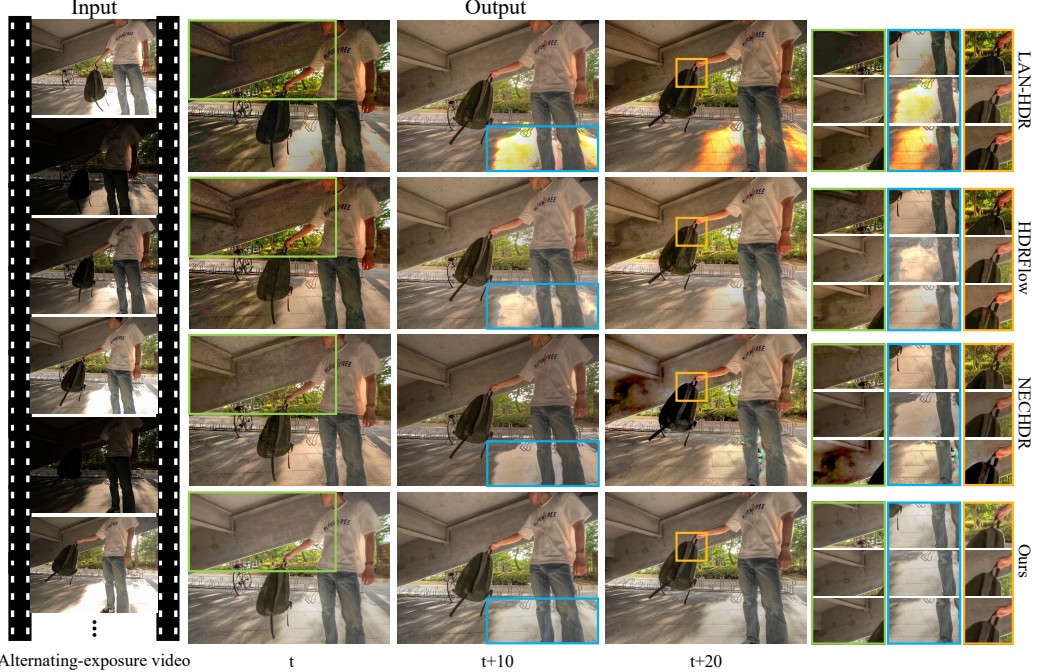

Figure 2: **Comparison of HDR video reconstruction on training views.** Given alternating-exposure video, HDR video reconstruction baselines, *i.e.*, LAN-HDR (Chung and Cho, 2023), HDRFlow (Xu et al., 2024), and NECHDR (Cui et al., 2024) fail to produce consistent results, while our model ensures temporal coherence and recovers valid information in saturated regions.

## 1 INTRODUCTION

Radiance of real-world scenes typically spans a much wider dynamic range than what standard digital sensors can capture. As a result, captures with standard cameras often suffer from overexposed highlights or underexposed shadows, leading to severe information loss in critical regions during the recording process. While high dynamic range (HDR) imaging aims to recover this lost information by merging alternating-exposure frames, conventional methods are fundamentally constrained to the 2D image plane. Existing video-based approaches (Kalantari et al., 2017; Chung and Cho, 2023; Xu et al., 2024; Cui et al., 2024) typically rely on 2D pixel-level alignment within a narrow temporal window of only 3 to 7 frames. Since these methods lack a physical understanding of the 3D scene, they often suffer from significant color drift and geometric flickering, as distant frames are not constrained to maintain radiometric and spatio-temporal consistency (see Fig. 2).

To address these limitations, we propose a paradigm shift from 2D pixel-based fusion to 4D spatio-temporal modeling. In this work, we introduce HDR-NSFF, a novel framework that jointly reconstructs HDR radiance, 3D scene flow, geometry, and tone-mapping from alternating-exposure monocular videos. Our method maps the entire video sequence into a single, unified 4D scene representation. This approach establishes a global temporal receptive field, ensuring that both appearance and geometry remain consistent across the entire duration of the video, regardless of the temporal distance between frames. By representing the scene as a continuous function of space and time, HDR-NSFF simplifies complex dynamics into physically plausible 3D scene flow. Our approach is built upon a 4D neural radiance field formulation and is compatible with emerging dynamic representations such as 4D Gaussian Splatting, making it broadly applicable across modern 4D scene reconstruction pipelines.

The reconstruction of dynamic HDR fields from monocular input is a highly ill-posed problem due to the coupled effects of exposure variance and limited-view observations. To handle these challenges, we build upon the 4D representation of Neural Scene Flow Fields (NSFF) (Li et al., 2021) and jointly optimize it with a learnable tone-mapping (TM) module. This allows the model to bridge the gap between varying LDR observations and the underlying HDR radiance in an end-to-end manner.

A critical challenge is that severe color inconsistency across frames degrades the reliability of standard motion priors (Teed and Deng, 2020). However, while pixel-level appearances fluctuate with exposure changes, the underlying semantic characteristics of an object remain invariant. Consequently, we utilize semantic features from DINOv2 (Oquab et al., 2023) to predict exposure-invariant dense flow, providing stable and consistent motion cues even under extreme exposure variations.

Another major challenge is the inherent information scarcity of alternating-exposure monocular videos, where observations are limited to a single viewpoint and frequently compromised by saturation. To address this, we incorporate a generative prior (Wu et al., 2025) as a regularizer to distill missing information into the radiance field. Instead of relying solely on degraded physical observations, our framework bootstraps the reconstruction by synthesizing enhanced multi-view supervision for unseen perspectives. This process allows HDR-NSFF to recover semantically plausible structures in regions where original pixels are completely lost, effectively mitigating the limited-view and saturation constraints of monocular capture.

To evaluate the proposed method, we construct the first real-world HDR-GoPro dataset. Captured with nine synchronized cameras, this dataset provides the first benchmark for dynamic HDR reconstruction with explicit multi-exposure variations across viewpoints. Extensive experiments on both real-world and synthetic data demonstrate that HDR-NSFF achieves state-of-the-art performance in novel view and time synthesis, recovering fine radiance details where previous methods (Martin-Brualla et al., 2021; Wu et al., 2024b; Zhu et al., 2024; Wu et al., 2024a) fail to produce consistent results. We further empirically validate the generality of our framework through additional experiments on a 4D Gaussian Splatting pipeline and 2D-to-4D HDR reconstruction baselines, confirming that our design is representation-agnostic (see Appendix for details). To summarize, our key contributions are:

- **4D HDR Framework:** We propose HDR-NSFF, shifting from 2D pixel-fusion to 4D spatio-temporal modeling to ensure global coherence and physical consistency across entire video.

- **Robust Learning Strategies:** We introduce exposure-robust motion estimation using semantic invariance of DINOv2 and employ generative priors as a regularizer to compensate for information loss in saturated, monocular views.

- **HDR-GoPro Dataset:** We provide the first real-world benchmark for cross-exposure space-time view synthesis, featuring nine synchronized cameras with explicit multi-exposure variations.

## 2 RELATED WORK

**High Dynamic Range Video Reconstruction**. High dynamic range (HDR) imaging from multi-exposure inputs has been extensively studied in computational photography. Early image-based approaches combine differently exposed photographs by estimating radiometric response functions and fusing measurements into HDR radiance maps (Debevec et al., 2023; Mitsunaga and Nayar, 1999). More recent works model HDR recovery using low-rank constraints across exposure stacks, enabling robustness to misalignment and outliers (Oh et al., 2013; 2014). Building upon these foundations, subsequent works extend HDR reconstruction to video by aligning and fusing alternating-exposure LDR frames (Kang et al., 2003; Kalantari et al., 2013; 2017; Chen et al., 2021; Chung and Cho, 2023; Xu et al., 2024; Cui et al., 2024). These approaches typically rely on optical flow or CNN-based alignment in 2D, followed by refinement to suppress ghosting. While effective for moderate motion, they remain vulnerable to occlusions, large displacements, and exposure-induced inconsistencies. In contrast, our work reconstructs HDR video in 4D, enabling consistent rendering.

**Dynamic Scene Reconstruction**. NeRF-based methods such as NSFF (Li et al., 2021), DynIBaR (Li et al., 2023), HyperNeRF (Park et al., 2021), and factorized grid models like HexPlane (Cao and Johnson, 2023) and K-Planes (Fridovich-Keil et al., 2023) have advanced free-viewpoint rendering of dynamic scenes. These methods represent a scene as a continuous function of space and time, sometimes augmented with deformation fields or canonical templates. They can synthesize novel views or even novel time steps. In parallel, 3D Gaussian Splatting has recently been extended to dynamic settings through 4DGS (Wu et al., 2024b), MotionGS (Zhu et al., 2024), Gaussian Marbles (Stearns et al., 2024), DeformableGS (Yang et al., 2024), and MoSca (Lei et al., 2025) achieving high efficiency and real-time rendering. Despite their success, all of these methods assume photometrically consistent LDR inputs and do not address the challenges of HDR content. Thus,

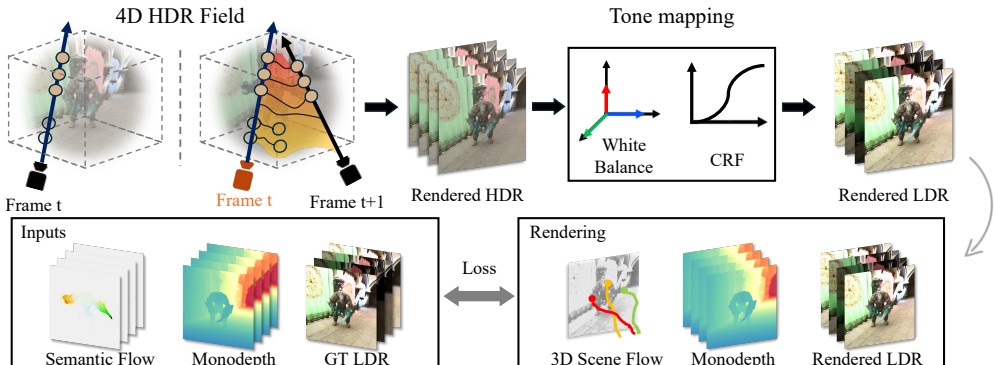

Figure 3: **Overall pipeline of our proposed method.** HDR-NSFF takes an alternating-exposure monocular video as input and estimates 3D scene flow for the sampled points along each ray. Neighboring frames are then warped to render the HDR radiance at the target frame, which is tone-mapped to LDR via a white-balance and camera-response function module. Photometric loss with the ground-truth LDR images, along with optical flow and depth constraints from off-the-shelf models, jointly optimize both the scene flow fields and tone-mapping module in an end-to-end manner.

they struggle to faithfully represent scenes with extreme lighting variations, whereas our approach explicitly targets HDR reconstruction of dynamic radiance fields.

**High Dynamic Range Novel View Synthesis**. Several recent works integrate HDR modeling into volumetric representations, mainly for static scenes. HDR-NeRF (Huang et al., 2022) and HDR-Plenoxel (Jun-Seong et al., 2022) model radiance together with tone-mapping or exposure functions, enabling HDR novel view synthesis from multi-exposure data. GaussHDR (Liu et al., 2025) extends HDR reconstruction to Gaussian Splatting with local tone mapping, while LTM-NeRF (Huang et al., 2024) embeds spatially varying tone mapping directly into NeRF. These works demonstrate the benefits of HDR-aware radiance fields but assume static content. The most relevant to our work is HDR-HexPlane (Wu et al., 2024a), which extends a factorized grid representation to dynamic HDR scenes by learning per-image exposure mappings. However, it does not explicitly model 3D motion, limiting its ability to represent complex dynamics and to perform temporal synthesis. In contrast, our method incorporates explicit motion modeling, allowing robust HDR reconstruction from real-world alternating-exposure videos and supporting both novel-view and novel-time rendering.

## 3 PRELIMINARY

**Neural Scene Flow Fields**.  Neural Scene Flow Fields (NSFF) (Li et al., 2021) extend NeRF (Mildenhall et al., 2021) to dynamic environments by representing a scene as a spatio-temporal radiance field. To handle moving objects, NSFF decomposes the scene into a static branch $F_\theta^{\text{st}}$ and a dynamic branch $F_\theta^{\text{dy}}$, where the latter is conditioned on time $t$. The core innovation of NSFF is the explicit modeling of 3D *scene flow* $\mathcal{F}_{t \to t \pm 1}$, which predicts the displacement of 3D points between adjacent frames:

$$(c_t^{\text{dy}}, \sigma_t^{\text{dy}}, \mathcal{F}_{t \to t \pm 1}, W_t) = F_\theta^{\text{dy}}(\mathbf{x}, \mathbf{d}, t). \tag{1}$$

This scene flow allows the model to warp 3D points across time, facilitating *temporal consistency constraints* in 3D space rather than 2D pixels. The static branch models time-invariant geometry, and both branches are fused via a blending weight $v$:

$$\hat{C}(\mathbf{r}, t) = \int_{z_n}^{z_f} T(z, t) \left[ v(z) c^{\text{st}}(z) \sigma^{\text{st}}(z) + (1 - v(z)) c_t^{\text{dy}}(z) \sigma_t^{\text{dy}}(z) \right] dz. \tag{2}$$

By optimizing these components jointly, NSFF reconstructs a continuous 4D representation that maintains geometric coherence over time, providing a robust foundation for our HDR reconstruction in dynamic settings. Details are provided in the Appendix B.

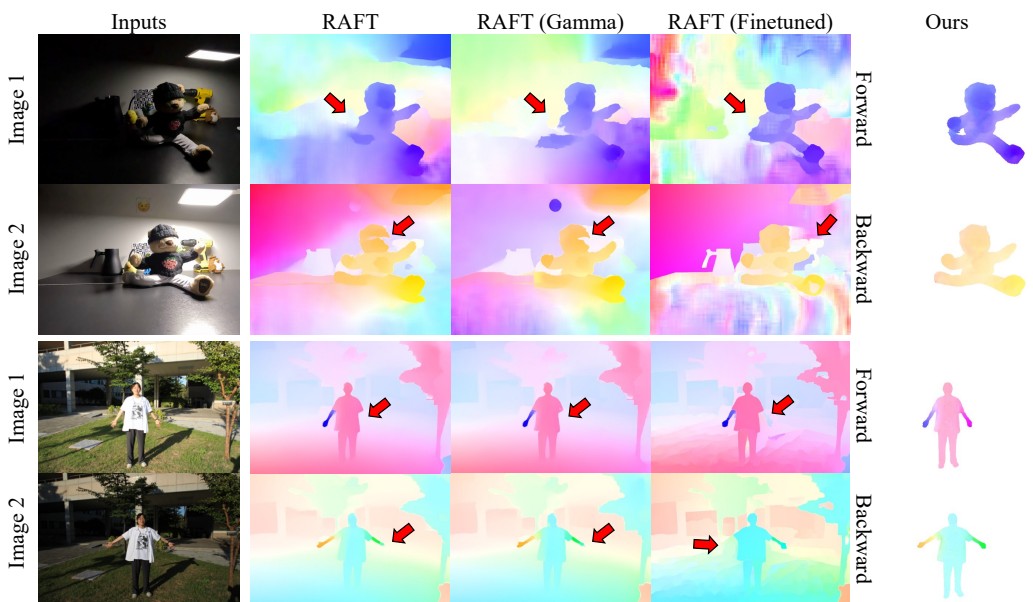

Figure 4: **Visualization of flow estimation under varying exposure conditions.** RAFT yields noticeable errors when exposures vary. Even with gamma correction, RAFT (Gamma) still fails to produce accurate results. While fine-tuning on synthetic data RAFT (Finetuned) shows moderate improvement, our proposed semantic-based approach achieves superior accuracy.

## 4 METHOD

HDR-NSFF reconstructs dynamic HDR radiance fields by jointly optimizing HDR radiance, 3D scene flow, geometry, and tone-mapping from alternating-exposure monocular videos. Building upon the 4D representation of NSFF (Sec. 3), our framework introduces a unified pipeline that disentangles appearance and motion under varying exposures. As illustrated in Fig. 3, HDR-NSFF integrates three core components to achieve physically plausible and temporally coherent HDR reconstruction: (i) joint optimization strategy incorporating a learnable tone-mapping module, (Sec. 4.1), (ii) exposure-robust semantic flow estimation for reliable motion learning (Sec. 4.2), (iii) generative prior regularization to compensate for the limited-view and saturation constraints of input video (Sec. 4.3).

### 4.1 TONE-MAPPING

A major challenge in HDR reconstruction is the need to model the nonlinear relationship between LDR observations and the underlying HDR radiance. To explicitly account for this nonlinearity, we introduce a learnable tone-mapping module $\mathcal{T}$ with radiometric parameter $\theta$ that maps rendered HDR radiance $E$ to the LDR domain:

$$C = \mathcal{T}(E; \theta) = g_\theta\big(w(E)\big), \tag{3}$$

where $w$ applies per-channel white balance correction and $g_\theta$ denotes the camera response function (CRF). To ensure stable optimization under extreme exposures, we employ a leaky-thresholded CRF that mitigates saturation effects, along with a smoothness regularization that encourages physically plausible CRF shapes through second-order derivative penalties. These components provide both flexibility and regularization, enabling $\mathcal{T}$ to form consistent HDR supervision across varying exposure levels and maintain a coherent radiance field in 4D space. Details are provided in the Appendix C.

### 4.2 SEMANTIC BASED OPTICAL FLOW

A key challenge in reconstructing HDR dynamic scenes from alternating-exposure video is that severe frame-to-frame color inconsistencies degrade the reliability of conventional optical flow methods (Teed and Deng, 2020) (see Fig. 4). We overcome this by leveraging the insight that while

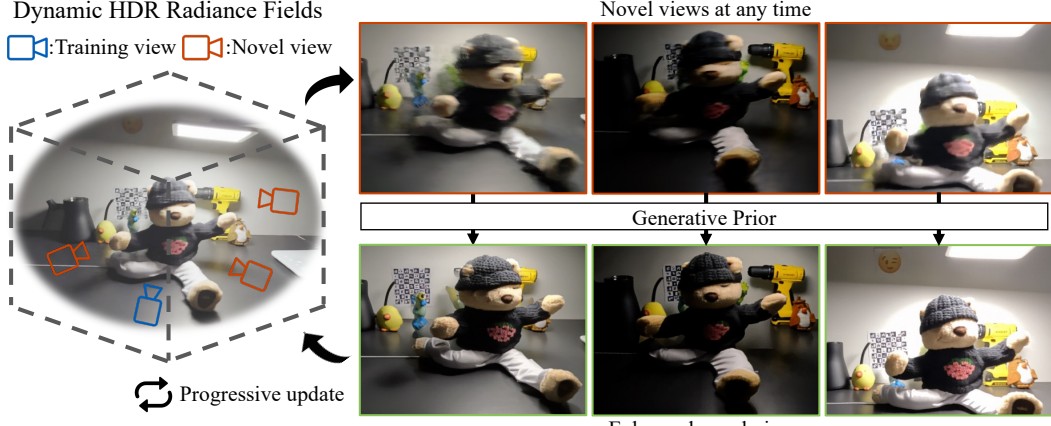

Figure 5: **Generative prior (GP) pipeline for HDR dynamic radiance fields optimization.** Unseen novel views are first rendered and then refined via a GP to restore details in regions with broken correspondences. These enhanced views serve as pseudo-labels for the progressive optimization of HDR dynamic radiance fields. This bootstrapping mechanism stabilizes correspondences, mitigating issues related to sparse viewpoints and saturated pixels in monocular videos with alternating exposures.

pixel-level appearances fluctuate with exposure changes, the semantic characteristics remain invariant. Consequently, we transition from color-dependent matching to semantic-based motion estimation.

In this context, we utilize the robust embedding space of DINOv2 (Oquab et al., 2023). Beyond its known resilience to photometric corruptions (Hendrycks and Dietterich, 2019), we empirically observe that DINOv2 features maintain consistency in varying exposure settings (see Appendix S2). Based on these observations, we adopt DINO-Tracker (Tumanyan et al., 2024) as our motion estimation backbone with task-specific modifications.

To prevent the accumulation of tracking errors over long sequences, we re-initialize tracking points at each timestep. This allows us to estimate precise bidirectional adjacent-frame flow as required by our pipeline. Furthermore, we incorporate motion masks from SAM2 (Ravi et al., 2024) to confine DINO-Tracker to dynamic regions, effectively filtering out tracking noise in the background. As a result, our semantic flow provides the stable and consistent motion cues (see Fig. 4).

### 4.3 GENERATIVE PRIOR AS REGULARIZER

Reconstructing dynamic HDR scenes from monocular videos is particularly challenging due to the inherent information scarcity. At any given timestamp, the scene is observed from only a single viewpoint, which is frequently compromised by saturation. Since these degraded physical observations cannot directly provide sufficient geometric and radiometric cues, we incorporate a generative prior (Wu et al., 2025) as a regularizer to distill missing information into the radiance field.

The core strategy is to bootstrap the 4D reconstruction by transforming the ill-posed monocular task into a pseudo-multi-view optimization problem. Specifically, our framework synthesizes enhanced supervision for unseen perspectives, enabling HDR-NSFF to recover semantically plausible structures in regions where original pixels are lost. During optimization, we periodically render a candidate novel viewpoint $\hat{C}$. We then obtain a generative enhancement $C^{\text{gen}} = \mathcal{G}(\hat{C}, C^{\text{ref}})$ by utilizing the generative prior $\mathcal{G}$, conditioned on the available training view $C^{\text{ref}}$ to maintain spatial consistency. $C^{\text{gen}}$ serve as pseudo observations, and we enforce alignment via a patch-wise perceptual loss:

$$\hat{\mathcal{L}}_{\text{gen}} = \sum_{p \in \mathcal{P}} \left\| \phi(\hat{C}_p) - \phi(C_p^{\text{gen}}) \right\|_1, \tag{4}$$

where $\phi$ denotes the perceptual encoder and $p$ indexes spatial patches. This regularization encourages the model to fill in saturated or unobserved regions with semantically consistent content. To prevent the prior from introducing hallucinations or overriding physical truths, we carefully control its influence through a scheduled activation. The generative loss is applied only after an initial warm-up

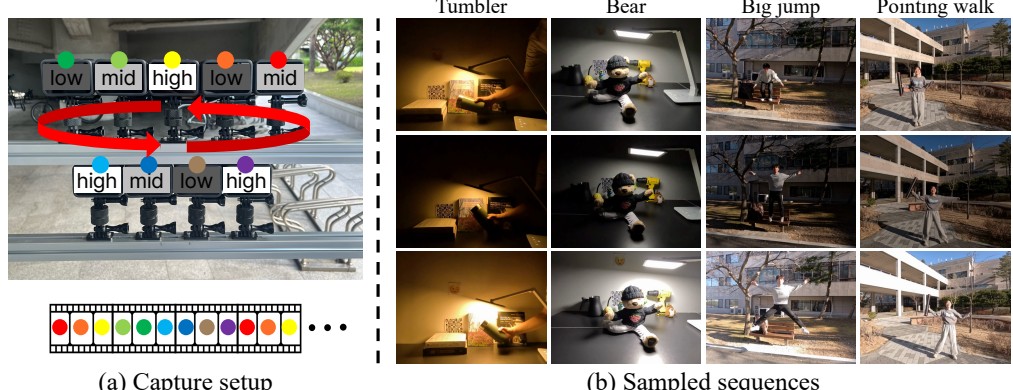

(a) Capture setup         (b) Sampled sequences

Figure 6: **Evaluation setup and sampled sequences from our proposed GoPro dataset**. To evaluate novel view synthesis, we use nine GoPro Hero 13 Black cameras synchronized to record multi-view video at three exposures (mid, low, high). We construct a alternating-exposure monocular video by selecting one frame per time step across exposures, and use the remaining views for evaluation.

period $T_{\text{warm}}$ to ensure the underlying geometry is reasonably established:

$$\alpha_{\text{gen}}(t) = \begin{cases} 0, & t < T_{\text{warm}} \\ p_{\text{gen}}, & t \geq T_{\text{warm}} \end{cases} \tag{5}$$

where $T_{\text{warm}}$ and sampling probability $p_{\text{gen}}$ are $200K$ and $0.1$, respectively. The generative loss is:

$$\mathcal{L}_{\text{gen}} = \alpha_{\text{gen}}(t)\,\beta_{\text{gen}}\,\hat{\mathcal{L}}_{\text{gen}}, \tag{6}$$

with $\beta_{\text{gen}}$ controlling the overall contribution as a regularizer. Figure 5 shows that this bootstrapping process effectively mitigates the limited-view and saturation constraints, ensuring geometric fidelity and radiance completeness across space and time. More details are provided in the Appendix C.

## 4.4 OBJECTIVE FUNCTION

We train both the neural scene flow fields and the tone-mapping module by minimizing the Mean Absolute Error (MAE) between rendered LDR views and ground-truth frames. Following NSFF (Li et al., 2021), we replace the rendered color $\hat{C}$ with our tone-mapped output $\mathcal{T}(\hat{E})$, where $\hat{E}$ denotes the rendered HDR radiance. The superscript cb denotes the combined rendering that fuses static and dynamic components of the scene. The photometric losses are:

$$\mathcal{L}_{\text{cb}} = \sum_{r_i} \|\mathcal{T}(\hat{E}_i^{\text{cb}}(r_i)) - C_i(r_i)\|_1, \quad \text{and} \tag{7}$$

$$\mathcal{L}_{\text{photo}} = \sum_{r_i} \sum_{j \in \mathcal{N}(i)} \|\mathcal{T}(\hat{E}_{j \to i}(r_i)) - C_i(r_i)\|_1, \tag{8}$$

where $r$ denotes a camera ray. Here, $\hat{E}_{j \to i}(r_i)$ denotes the HDR radiance warped from a frame $j$ to $i$. We also adopt the optical flow and single-view depth prior, denoted $\mathcal{L}_{\text{Flow}}$ and $\mathcal{L}_{\text{depth}}$ to regularize monocular reconstruction followed by NSFF (Li et al., 2021). For the CRF and generative prior objective functions, we apply $\mathcal{L}_{\text{smooth}}$ and $\mathcal{L}_{\text{gen}}$, respectively. The total objective function of our HDR-NSFF is as follows:

$$\mathcal{L} = \mathcal{L}_{\text{cb}} + \mathcal{L}_{\text{photo}} + \beta_{\text{data}}\mathcal{L}_{\text{data}} + \beta_{\text{reg}}\mathcal{L}_{\text{reg}} + \beta_{\text{smooth}}\mathcal{L}_{\text{smooth}} + \mathcal{L}_{\text{gen}}, \tag{9}$$

where $\beta$ are coefficients weight each term. The details can be found in Appendix B, C.

## 4.5 DATASETS

**Proposed HDR-GoPro Dataset**. To evaluate HDR space-time view synthesis, we construct the first real-world dataset using nine synchronized GoPro cameras arranged in a nearly parallel configuration. The cameras are divided into three groups, each assigned to a different exposure level (low, mid, high). This setup allows us to evaluate whether the model, trained on alternating-exposure frames, can successfully reconstruct the underlying radiance and synthesize views at any target exposure. The dataset comprises 12 diverse indoor and outdoor scenes featuring complex motions (see Fig. 6).

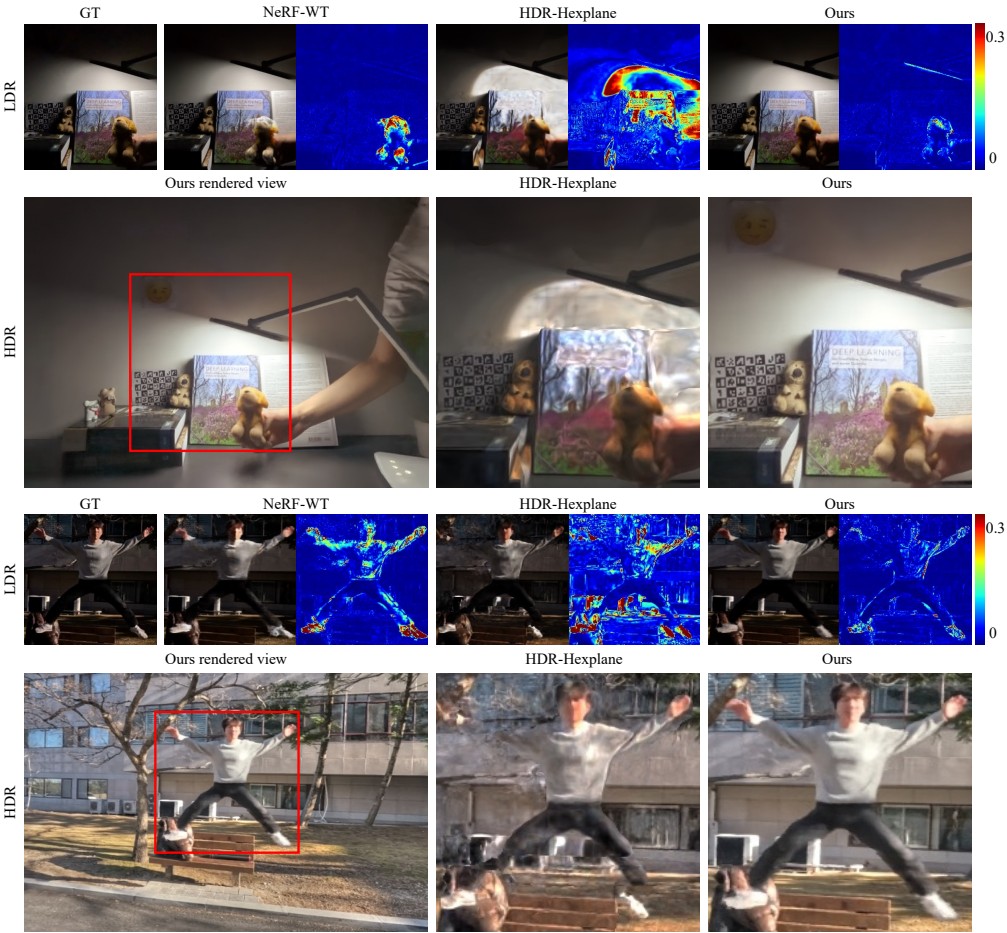

Figure 7: **Qualitative results of novel view synthesis on GoPro dataset.** The odd-numbered rows show the LDR-rendered novel views along with their corresponding L1 error maps against the ground-truth novel view (leftmost). Our method consistently yields the smallest error across all scenes. The even-numbered rows present tone-mapped HDR novel views. Compared with HDR-HexPlane (Wu et al., 2024a), our approach produces more accurate radiance, geometry, and motion representations.

## 5 EXPERIMENTS

In this section, we evaluate the effectiveness of HDR-NSFF in reconstructing high-quality 4D HDR scenes from alternating-exposure monocular videos. We first present the primary results on novel view and time synthesis using both synthetic and our newly constructed real-world HDR-GoPro datasets (Sec. 5.1). To validate the fidelity of the reconstructed radiance, we also provide a qualitative comparison with ground-truth HDR images and histogram analyses. Then, we conduct an in-depth analysis of our robust learning strategies (Sec. 5.2), where we justify our design choices through controlled experiments under extreme exposure variations. For all HDR visualizations, we employ a consistent Photomatix Pro tone-mapping operator to ensure a fair qualitative comparison.

### 5.1 RESULTS

**Novel view synthesis**. We evaluate novel view synthesis on our proposed HDR-GoPro dataset. At each timestamp, we use a single camera's frame for training and reserve the remaining eight viewpoints, which capture the same moment at different exposures as ground truth. During testing, the reconstructed HDR radiance is tone-mapped to the target exposure for quantitative comparison (e.g., PSNR, SSIM, LPIPS) against these LDR references. This ensures a rigorous assessment of both temporal coherence and exposure-invariant geometric reconstruction. Table 1 shows that our approach

| Methods | Full | | | Dynamic only | | |
|---|---|---|---|---|---|---|
| | PSNR↑ | SSIM↑ | LPIPS↓ | PSNR↑ | SSIM↑ | LPIPS↓ |
| NSFF | 18.02 | 0.6792 | 0.2061 | 17.59 | 0.5473 | 0.2329 |
| 4DGS | 20.94 | 0.7905 | 0.1541 | 17.83 | 0.5524 | 0.2230 |
| MotionGS | 14.61 | 0.3976 | 0.3617 | 12.33 | 0.2303 | 0.4696 |
| NeRF-WT | 29.70 | 0.9333 | 0.0598 | 19.25 | 0.6335 | 0.1770 |
| HDR-HexPlane | 20.70 | 0.6694 | 0.1917 | 20.55 | 0.6629 | 0.1716 |
| Ours (w/o GP & DT) | 31.04 | 0.9364 | 0.0621 | 24.93 | 0.8068 | 0.1048 |
| Ours (w/o GP) | 32.66 | 0.9447 | 0.0557 | 25.65 | 0.8205 | 0.1012 |
| Ours | 32.63 | 0.9444 | 0.0554 | 25.50 | 0.8208 | 0.0972 |

Table 1: **Averaged quantitative results of novel view synthesis on GoPro dataset.** Ours achieves the best overall performance, with DINO-Tracker (DT) offering the strongest improvement in motion-consistent reconstruction and the generative prior (GP) further enhancing perceptual quality.

| Methods | Full | | | Dynamic only | | |
|---|---|---|---|---|---|---|
| | PSNR↑ | SSIM↑ | LPIPS↓ | PSNR↑ | SSIM↑ | LPIPS↓ |
| NSFF | 19.01 | 0.7258 | 0.1976 | 18.84 | 0.5873 | 0.2531 |
| HDR-HexPlane | 20.46 | 0.6583 | 0.1933 | 19.59 | 0.6107 | 0.1855 |
| Ours (w/o GP & DT) | 31.31 | 0.9392 | 0.0648 | 24.85 | 0.7979 | 0.1372 |
| Ours (w/o GP) | 32.79 | 0.9451 | 0.0596 | 25.40 | 0.8075 | 0.1378 |
| Ours | 32.75 | 0.9448 | 0.0594 | 25.26 | 0.8070 | 0.1339 |

Table 2: **Averaged quantitative results of novel view and time synthesis on GoPro dataset.** Our method outperforms baseline models.

| Methods | Full | | | Dynamic only | | |
|---|---|---|---|---|---|---|
| | PSNR↑ | SSIM↑ | LPIPS↓ | PSNR↑ | SSIM↑ | LPIPS↓ |
| NSFF | 15.98 | 0.6457 | 0.1388 | 16.04 | 0.5697 | 0.1527 |
| NeRF-WT | 31.10 | 0.9366 | 0.0342 | 21.50 | 0.7490 | 0.0895 |
| HDR-HexPlane | 29.95 | 0.9055 | 0.0527 | 23.87 | 0.7999 | 0.1071 |
| Ours | 35.07 | 0.9465 | 0.0483 | 27.19 | 0.8836 | 0.0576 |

Table 3: **Averaged quantitative results of novel view and time synthesis on synthetic data.** Our method outperforms baseline models.

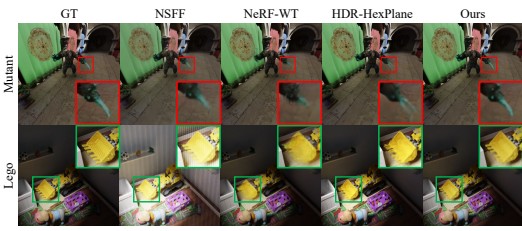

Figure 8: **Qualitative results of novel view and time synthesis on synthetic data.** Since our approach explicitly models scene flow, it excels at time interpolation.

achieves significant improvements in rendering fidelity compared to baselines, both in highly dynamic regions and across the entire scene. Figure 7 shows its effectiveness in reconstructing HDR scenes with fine detail across varying exposures. Methods without appearance embedding (NSFF (Li et al., 2021), 4DGS (Wu et al., 2024b), MotionGS (Zhu et al., 2024)) fail to reconstruct consistent HDR views under alternating exposures. NeRF-WT (Quei-An, 2020) and HDR-Hexplane (Wu et al., 2024a) provide limited robustness, but still struggle in real-world settings.

**Novel view and time synthesis**. We further evaluate novel view and time synthesis to assess the model's ability to interpolate dynamic motion from temporally sparse observations (see Fig. 8). We train the model using only the even-indexed frames of the input video, effectively doubling the temporal gap between training samples. For evaluation, we then synthesize the held-out odd-indexed frames from unseen camera viewpoints. This task requires the model to simultaneously interpolate 3D scene flow for temporal consistency and synthesize novel spatial views. As shown in Table 3, our method consistently outperforms competing models across all metrics, demonstrating superior robustness in handling complex dynamics even with reduced temporal sampling.

We also conduct this joint evaluation on our real-world HDR-GoPro dataset by synthesizing scenes at held-out time instances and camera viewpoints. As shown in Table 2, our model consistently surpasses all baseline methods, confirming its ability to handle complex real-world dynamics with high radiometric and geometric accuracy. The success of HDR-NSFF stems from its explicit modeling of 3D scene flow, which enables reliable synthesis across both space and time. In contrast, grid-based representations like HDR-HexPlane (Wu et al., 2024a) lack explicit motion modeling, which inherently limits their capacity for accurate space-time interpolation in dynamic HDR scenes.

**Qualitative comparison of HDR reconstruction**. To validate our HDR reconstruction, we qualitatively compare our results with ground-truth HDR images (see Fig. 9). Tone-mapped HDR views from our model closely match ground truth, preserving fine details in both under- and overexposed regions. Histograms of pixel intensities further show that our reconstructions cover the full radiance range, recovering values from very low to high intensities. In addition, novel LDR views rendered at multiple exposures confirm that our method accurately controls exposure.

## 5.2 EXPOSURE ROBUST LEARNING STRATEGIES

**Depth analysis under varying exposure.** We assess the robustness of off-the-shelf depth estimators under exposure variation by synthetically generating ±2 EV versions of input images. RGB images are first converted into pseudo-RAW using a learned ISP inversion

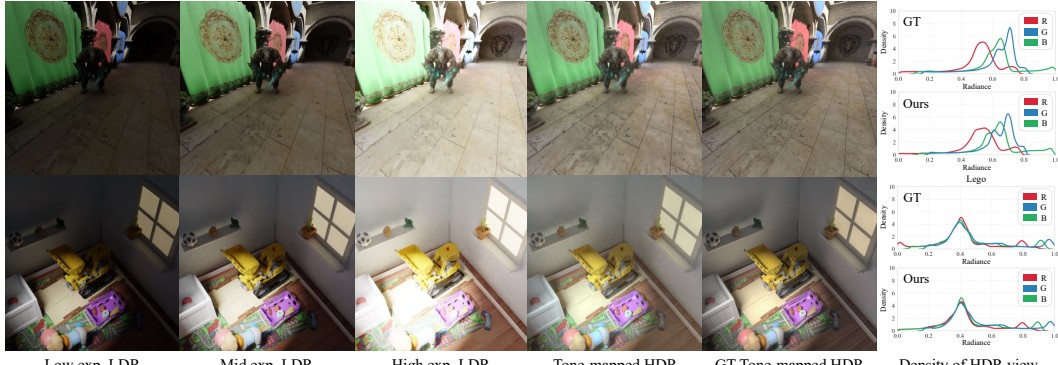

|  |  |  |  |  |  |
|---|---|---|---|---|---|
| Low exp. LDR | Mid exp. LDR | High exp. LDR | Tone-mapped HDR | GT Tone-mapped HDR | Density of HDR view |

Figure 9: **Qualitative results on novel LDR/HDR view synthesis.** We visualize LDR rendering results at varying exposure levels (low, mid, and high), tone-mapped HDR rendering by ours and corresponding ground-truth HDR references. We also visualize histograms of our HDR images and ground truth. For better visualization, we plot HDR histogram using smoothed KDE method.

model (Xing et al., 2021), after which ±2 EV renderings are produced via standard sRGB mapping. Following the protocol of Ke et al. (Ke et al., 2024), we evaluate AbsRel on NYUv2 and ScanNet (Silberman et al., 2012; Dai et al., 2017). As shown in Table 4, while all methods degrade under ±2 EV shifts, Depth-Anything-V2 remains significantly more robust than alternatives. Based on this observation, we adopt Depth-Anything-V2 as the geometric prior in our pipeline.

| Methods | NYUv2 | | | ScanNet | | |
|---|---|---|---|---|---|---|
|  | Original | +2EV | -2EV | Original | +2EV | -2EV |
| MiDaS | 9.08 | 13.68 | 9.35 | 8.66 | 13.78 | 10.22 |
| DPT | 9.21 | 12.96 | 8.95 | 8.27 | 13.62 | 9.57 |
| Marigold | 5.81 | 11.26 | 6.66 | 7.24 | 14.26 | 8.33 |
| Depth-Anything-V2 | 4.87 | 7.63 | 5.10 | 4.82 | 10.57 | 6.36 |

Table 4: **Depth estimation results under exposure variance.** We employ AbsRel as the evaluation metric.

**Tone-mapping module analysis**. Prior HDR radiance studies explored three CRF designs: a non-learnable fixed CRF (Wu et al., 2024a), a fully learnable MLP-based CRF (Huang et al., 2022), and a piecewise parametric CRF with per-channel white-balance factors (Jun-Seong et al., 2022).

| Methods | Full | | | Dynamic only | | |
|---|---|---|---|---|---|---|
|  | PSNR↑ | SSIM↑ | LPIPS↓ | PSNR↑ | SSIM↑ | LPIPS↓ |
| w/o Tone-mapping | 17.79 | 0.7048 | 0.0705 | 15.59 | 0.5577 | 0.1339 |
| Fix CRF | 25.55 | 0.8391 | 0.0487 | 20.43 | 0.6904 | 0.0911 |
| MLP CRF | 28.76 | 0.8861 | 0.0394 | 21.48 | 0.7256 | 0.0776 |
| Piecewise CRF | 31.01 | 0.9301 | 0.0233 | 22.55 | 0.7714 | 0.0697 |

Table 5: **Comparison of tone-mapping designs.**

The fixed CRF offers strong regularization but insufficient flexibility, whereas the MLP CRF is overly flexible and often unstable. We adopt the piecewise CRF, which provides a balanced formulation. On our HDR-GoPro dataset, piecewise CRF achieves the best novel-view synthesis performance (Table 5), indicating that moderate flexibility with structured regularization is most effective under varying exposures.

Beyond our main experiments, we validate the generality of our framework by integrating our modules into a 4D Gaussian Splatting (4DGS) pipeline (Lei et al., 2025) and comparing against 2D-to-4D HDR reconstruction baselines. Detailed results are provided in the Appendix E, F.

## 6 CONCLUSION

In this work, we introduced HDR-NSFF, the first framework to jointly reconstruct HDR radiance, 3D scene flow, geometry, and tone-mapping from alternating-exposure monocular videos. By shifting the paradigm from 2D pixel-based fusion to 4D spatio-temporal modeling, our approach overcomes the fundamental limitations of prior methods, such as color drift and geometric flickering. We leveraged the semantic invariance of DINOv2 for robust motion estimation and incorporated 3D-aware generative priors as a regularizer to recover information in saturated regions. Extensive experiments on synthetic and our newly proposed HDR-GoPro dataset demonstrate that HDR-NSFF consistently outperforms state-of-the-art baselines in novel view and time synthesis.

**Limitations**. Our method relies on pre-computed camera pose by COLMAP (Schonberger and Frahm, 2016), which can be difficult to estimate in extreme exposure-varying scenes. Furthermore, our current model does not explicitly handle motion blur from long exposures. Addressing these efficiency and robustness challenges remains an important direction for future 4D HDR research.

## ACKNOWLEDGMENTS

We thank Lee Jung-Mok, GeonU Kim, and Donghyun Kim for their assistance in data acquisition, and the members of AMILab for helpful discussions.

This work was supported by Institute of Information & communications Technology Planning & Evaluation (IITP) grant funded by the Korea government (MSIT) (No.RS-2025-25443318, Physically-grounded Intelligence: A Dual Competency Approach to Embodied AGI through Constructing and Reasoning in the Real World), Institute of Information & communications Technology Planning & Evaluation (IITP) grant funded by the Korea government (MSIT) (No. RS-2024-00457882, National AI Research Lab Project), the National Research Foundation of Korea (NRF) grant funded by the Korea government (MSIT) (No. RS-2024-00358135, Corner Vision: Learning to Look Around the Corner through Multi-modal Signals), and the InnoCORE program of the Ministry of Science and ICT (25-InnoCORE-01).

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

## APPENDIX

This appendix provides additional technical details and experimental results complementary to the main manuscript. Section A describes the implementation details, including hardware configuration and training setups. Section B elaborates on the Neural Scene Flow Fields (NSFF) framework and its regularization terms, while Section C details our HDR-specific components, including tone-mapping, semantic tracking, and the generative prior formulation.

Additional experimental results and ablation studies are presented in Section D. We further demonstrate the general utility of our modules through an extension to 3DGS-based reconstruction in Section E, followed by a comparative analysis against 2D-to-4D HDR reconstruction pipelines in Section F. Finally, Section G provides extensive qualitative results on the HDR-GoPro dataset. We highly recommend watching the supplementary video for a better assessment of the temporal coherence and dynamic HDR rendering quality of our results.

## CONTENTS

## A    IMPLEMENTATION DETAILS

### A.1    COUNTERPARTS

In this chapter, we briefly explain the method we compared as a counterpart in our experiments.

**NeRF-WT**. NeRF-W (Martin-Brualla et al., 2021) introduces per-image appearance and transient embedding, modeling to handle dynamic changes such as lighting variations and moving objects. In our experiments, we adapted NeRF-W to a dynamic HDR video (named NeRF-WT) using appearance embedding for ISP modeling and transient part for scene dynamics. We follow the hyperparameters given in the codebase. For implementation we used the codebase in `https://github.com/kwea123/nerf_pl`

**HDR-Hexplane**. HDR-Hexplane (Wu et al., 2024a) adopted Hexplane (Cao and Johnson, 2023) for the dynamic 3D representation and MLP with exposure embeddings accompanied with fixed gamma function to optimize ISP module. We follow the hyperparameters following manuscript. For implementation we used the codebase in `https://github.com/hustvl/HDR-HexPlane`

### A.2    DATASET

**Synthetic**. We select four synthetic scenes for evaluation: *Lego*, *Mutant*, *Jumping Jack*, and *Stand Up*. Each image has a resolution of $800 \times 800$, with exposure values spanning from -2EV to 5EV. To maximize the influence of exposure change, we carefully adjust the camera viewpoints and lighting directions.

The sampling rate is determined based on the motion speed of each scene. Specifically, the *Lego* scene is subsampled by selecting every 10th frame, whereas the remaining scenes are sampled by skipping every two frames.

**Real**. For the real dataset, we preset exposure time for each camera before acquisition. We set exposure time differently for each sequence. Sequence lengths and corresponding exposure information are detailed in the Table S1 All sequences are synchronized using the GoPro software.

| Name | Exp. Time [s] | # of frames |
|---|---|---|
| Big jump | $\frac{1}{960}, \frac{1}{2880}, \frac{1}{7680}$ | 324 |
| Side walk | $\frac{1}{960}, \frac{1}{2880}, \frac{1}{7680}$ | 324 |
| Jumping jack | $\frac{1}{720}, \frac{1}{1920}, \frac{1}{7680}$ | 324 |
| Pointing walk | $\frac{1}{720}, \frac{1}{1920}, \frac{1}{7680}$ | 324 |
| Tube toss | $\frac{1}{720}, \frac{1}{1920}, \frac{1}{7680}$ | 324 |
| Bear | $\frac{1}{120}, \frac{1}{480}, \frac{1}{1920}$ | 324 |
| Dog | $\frac{1}{120}, \frac{1}{480}, \frac{1}{1920}$ | 324 |
| Tumbler | $\frac{1}{120}, \frac{1}{480}, \frac{1}{1920}$ | 324 |
| Fire extinguisher | $\frac{1}{480}, \frac{1}{960}, \frac{1}{1920}$ | 324 |
| Laptop | $\frac{1}{480}, \frac{1}{960}, \frac{1}{1920}$ | 324 |
| Bag | $\frac{1}{120}, \frac{1}{480}, \frac{1}{1920}$ | 324 |
| Ball | $\frac{1}{120}, \frac{1}{480}, \frac{1}{1920}$ | 324 |

Table S1: **Parameter settings for real dataset.**

## A.3 EXPERIMENTAL SETUP

To facilitate understanding of the experimental setup employed for the real dataset experiments, we provide an illustrative diagram in Fig. S1 In the novel view synthesis experiment, performance is evaluated by measuring the differences between synthesized results and the images captured from cameras that were excluded from the training set, for all camera views $i$. In the novel view and time synthesis experiment, we evaluate performance by holding out certain segments of the time sequence and measuring how accurately these withheld segments are inferred.

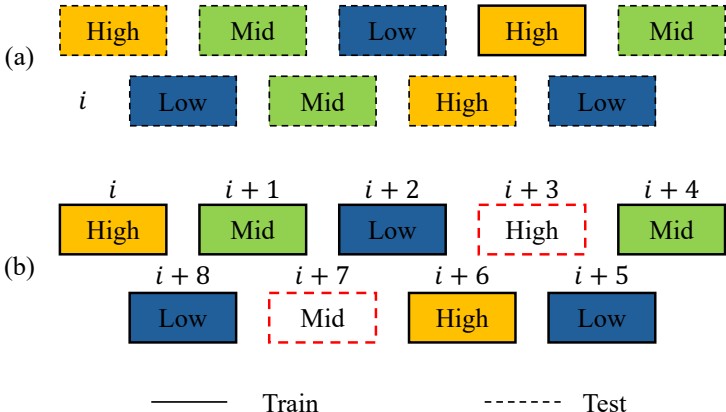

Figure S1: **Illustration of two experimental settings.** We illustrate two experimental settings described in Sec. 4.2 in manuscript: (a) Novel view synthesis (b) Novel view and time synthesis.

## B DETAILS OF NEURAL SCENE FLOW FIELDS

To model dynamic scenes, NSFF (Li et al., 2021) extend the concept of NeRF (Mildenhall et al., 2021) by representing 3D motion as scene flow fields. NSFF learns a combination of static and dynamic NeRF representations. The dynamic model, denoted as $F_\theta^{\text{dy}}$, explicitly models view and time dependent variations by incorporating time $t$ as an additional input. Beyond color and density, it also predicts forward and backward 3D scene flow $F_t = (\mathbf{f}_{t \to t+1}, \mathbf{f}_{t \to t-1})$ and occlusion weights $W_t = (w_{t \to t+1}, w_{t \to t-1})$ to handle 3D motion disocclusion:

$$(c_t, \sigma_t, F_t, W_t) = F_\theta^{\text{dy}}(\mathbf{x}, \mathbf{d}, t). \tag{10}$$

To supervise scene flow estimation, NSFF uses temporal photometric consistency. Specifically, for each time $i$, scene flow is predicted for the 3D points sampled along rays, and this predicted flow is

used to warp corresponding points from neighboring times $j \in \mathcal{N}(i)$ to time $i$. The color and opacity information of the warped points is then used to render the image at time $i$:

$$\hat{C}_{j\to i}(r_i) = \int_{z_n}^{z_f} T_j(z)\,\sigma_j\big(r_{i\to j}(z)\big)\,c_j\big(r_{i\to j}(z), d_i\big)\,dz, \tag{11}$$

$$\text{where} \quad r_{i\to j}(z) = r_i(z) + \mathbf{f}_{i\to j}\big(r_i(z)\big). \tag{12}$$

Temporal photometric consistency is enforced by minimizing the mean squared error (MSE) between the warped rendered view and the ground-truth image:

$$\mathcal{L}_{\text{photo}} = \sum_{r_i}\sum_{j\in\mathcal{N}(i)} \|\hat{C}_{j\to i}(r_i) - C_i(r_i)\|_2^2. \tag{13}$$

The static NeRF, $F_\theta^{\text{st}}$, represents a time-invariant scene using a multilayer perceptron (MLP). Given an input position $\mathbf{x}$ and view direction $\mathbf{d}$, it outputs the RGB color $c$, volume density $\sigma$, and an unsupervised 3D mixing weight $v$ that determines the blending between static and dynamic components:

$$(c, \sigma, v) = F_\theta^{\text{st}}(\mathbf{x}, \mathbf{d}). \tag{14}$$

Here, $c_t$ and $\sigma_t$ denote the color and volume density at position $\mathbf{x}$ at time $t$. The final color is computed by blending the static and dynamic components using the following rendering equation:

$$\hat{C}_i^{cb}(r_i) = \int_{z_n}^{z_f} T_i^{cb}(z)\,\sigma_i^{cb}(z)\,e_i^{cb}(z)\,dz, \tag{15}$$

where $\sigma_i^{cb}(z)c_i^{cb}(z)$ is a linear combination of static and dynamic scene components, weighted by $v(z)$:

$$\sigma_i^{cb}(z)c_i^{cb}(z) = v(z)c(z)\sigma(z) + (1 - v(z))c_i(z)\sigma_i(z). \tag{16}$$

$T_i$ represents the transmittance at time $i$, while $z_n$ and $z_f$ denote the near and far depths along the ray. The final rendered output $\hat{C}_i^{cb}(r_i)$ is optimized against the ground-truth pixel color $C_i(r_i)$ using a photometric loss:

$$\mathcal{L}_{cb} = \sum_{r_i} \|\hat{C}_i^{cb}(r_i) - C_i(r_i)\|_2^2. \tag{17}$$

Reconstructing dynamic scenes from monocular input is inherently ill-posed, and relying solely on photometric consistency often leads to convergence at poor local minima. Therefore, NSFF incorporates three additional guided losses: a term enforcing monocular depth and optical flow consistency, a motion trajectory term promoting cycle-consistency and spatiotemporal smoothness, and a compactness prior encouraging binary scene decomposition and reducing floaters via entropy and distortion losses.

Following section, we elaborate on data-driven prior loss (Flow loss and Single-view depth loss) and additional regularization terms introduced by NSFF (Li et al., 2021): Scene Flow Cycle Consistency and Low-Level regularization term. We employ additional regularization terms consistently in both our model and NSFF.

**Flow Loss**. Flow loss operates by minimizing the discrepancy between observed 2D pixel correspondences, computed from pretrained optical flow networks and predicted 2D pixel correspondences, obtained by projecting predicted 3D scene flows. This aligns 3D scene flow with pretrained 2D motion estimation.

Given two adjacent frames at times $i$ and $j = i \pm 1$, $L_{flow}$ is calculated as follows. Let $p_i$ represent a pixel location at frame $i$. The corresponding pixel location at frame $j$, denoted by $p_{i\to j}$, can be computed using pretrained 2D motion estimation $u_{i\to j}$ as $p_{i\to j} = p_i + u_{i\to j}$.

The model predicts an expected scene flow $\hat{F}_{i\to j}(r_i)$ corresponding to 3D location $\hat{X}_i(r_i)$ along the ray $r_i$ passing through the pixel $p_i$ via volumetric rendering. Thus, the predicted 3D displacement can be expressed as $\hat{X}_i(r_i) + \hat{F}_{i\to j}(r_i)$. Then, by applying the perspective projection operator $\Pi_j$, corresponding to the camera viewpoint at frame $j$, the expected 2D pixel position $\hat{p}_{i\to j}(r_i)$ at frame $j$ is calculated as:

$$\hat{p}_{i\to j}(r_i) = \Pi_j\left(\hat{X}_i(r_i) + \hat{F}_{i\to j}(r_i)\right). \tag{18}$$

Finally, the geometric consistency loss is computed by measuring the discrepancy between these two pixel positions (observed vs. predicted) using the L1-norm:

$$\mathcal{L}_{\text{flow}} = \sum_{r_i} \sum_{j \in \{i \pm 1\}} ||\hat{p}_{i \to j}(r_i) - p_{i \to j}(r_i)||_1. \tag{19}$$

**Single-view Depth Prior**. Encourages rendered depths to match predictions from a pretrained depth model:

$$\mathcal{L}_{\text{depth}} = \sum_{r_i} ||\hat{Z}_i^*(r_i) - Z_i^*(r_i)||_1, \tag{20}$$

where the superscript $(*)$ denotes scale-shift invariant normalization. These priors are combined into:

$$\mathcal{L}_{\text{data}} = \mathcal{L}_{\text{flow}} + \beta_{\text{depth}} \mathcal{L}_{\text{depth}}. \tag{21}$$

**Scene Flow Cycle Consistency**. To ensure plausible scene motion, the loss ensures coherence between forward and backward predicted scene flows for adjacent frames, mathematically defined as:

$$\mathcal{L}_{\text{cyc}} = \sum_{x_i} \sum_{j \in \{i \pm 1\}} w_{i \to j} ||f_{i \to j}(x_i) + f_{j \to i}(x_{i \to j})||_1, \tag{22}$$

where $f_{i \to j}(x_i)$ indicates the predicted displacement (scene flow) of point $x_i$ from time $i$ to $j$.

**Low-Level Regularization**. Spatial-temporal smoothness is enforced through $l1$ regularization on scene flow estimated between neighboring sampled 3D points along rays. This encourages 3D point trajectories to be piecewise linear. Another sparsity regularization term, calculating an $l1$ loss in flow estimation is also applied. This encourage minimal scene flow magnitudes across most spatial regions. It is composed of three equally weighted components: spatial smoothness, temporal smoothness, and minimal flow magnitude:

$$\mathcal{L}_{\text{reg}} = \mathcal{L}_{\text{sp}} + \mathcal{L}_{\text{temp}} + \mathcal{L}_{\text{min}}.$$

**Spatial Smoothness.** Following NSFF, the spatial smoothness term encourages nearby 3D samples along the same camera ray to predict similar scene flows. For each sampled 3D location $\mathbf{x}_i$ on ray $\mathbf{r}_i$, we consider its neighboring samples $\mathcal{N}(\mathbf{x}_i)$ and penalize the weighted $\ell_1$ discrepancy:

$$\mathcal{L}_{\text{sp}} = \sum_{\mathbf{x}_i} \sum_{\mathbf{y}_i \in \mathcal{N}(\mathbf{x}_i)} \sum_{j \in \{i \pm 1\}} w^{\text{dist}}(\mathbf{x}_i, \mathbf{y}_i) ||f_{i \to j}(\mathbf{x}_i) - f_{i \to j}(\mathbf{y}_i)||_1, \tag{1}$$

where the weight is based on Euclidean distance: $w^{\text{dist}}(\mathbf{x}, \mathbf{y}) = \exp(-2||\mathbf{x} - \mathbf{y}||_2)$.

**Temporal Smoothness.** The temporal term encourages each 3D trajectory to maintain low kinetic energy. This is implemented by minimizing the squared norm of the sum of forward and backward scene flows:

$$\mathcal{L}_{\text{temp}} = \frac{1}{2} \sum_{\mathbf{x}_i} ||f_{i \to i+1}(\mathbf{x}_i) + f_{i \to i-1}(\mathbf{x}_i)||_2^2. \tag{2}$$

**Minimal Flow Prior.** Finally, following the observation that most points in the scene exhibit small motion, we impose an $\ell_1$ penalty on all predicted scene flows to encourage near-zero flows where appropriate:

$$\mathcal{L}_{\text{min}} = \sum_{\mathbf{x}_i} \sum_{j \in \{i \pm 1\}} ||f_{i \to j}(\mathbf{x}_i)||_1. \tag{3}$$

## C    DETAILS OF HIGH DYNAMIC RANGE NEURAL SCENE FLOW FIELDS

Our method is built upon the NSFF framework and therefore inherits its core loss formulation and optimization structure. However, reconstructing HDR dynamic radiance fields from alternating-exposure monocular videos introduces unique challenges not addressed in the original NSFF design. To handle severe exposure fluctuations, saturation artifacts, and inconsistent appearance across viewpoints, we adapt NSFF's formulation by modifying the photometric loss (Eq. 8) and combined loss (Eq. 7), and by introducing additional regularizers tailored for HDR reconstruction. Specifically, our HDR-NSFF incorporates (i) a physically informed tone-mapping module, and (ii) a generative prior that recovers saturated or missing information. We describe each of these components in detail below, along with the updated objective function used for end-to-end optimization.

### C.1 TONE-MAPPING

Our goal is to reconstruct HDR dynamic radiance fields, encompassing both 3D space and motion, from 2D multi-exposure LDR RGB images. A crucial component in this process is the tone-mapping module, which bridges the gap between varying 2D observations and a coherent 3D HDR representation. Specifically, tone-mapping module, $\mathcal{T}$ can be expressed as:

$$C = \mathcal{T}(E, \theta) = g\big(w(E)\big), \tag{23}$$

where $E$ denotes the rendered radiance, $w$ the white balance correction, $g$ the camera response function (CRF), and $\theta$ the radiometric parameters.

The white balance function $w$ applies per-channel scaling using the white balance parameter $\theta_w = [w_r, w_g, w_b]^\top \in \mathbb{R}^3$, producing a white balance-corrected image $E_w$. The CRF $g$ is then applied to $E_w$, mapping it to the final LDR image $C$. The CRF is parameterized as a piecewise linear function, defined using 256 points uniformly sampled in the $[0, 1]$ range. Values exceeding the dynamic range are thresholded accordingly. During training, we adopt leaky-thresholding, to reduce saturation loss in rendered images:

$$g_{\text{leaky}}(x) = \begin{cases} \alpha x, & x < 0 \\ g(x), & 0 \le x \le 1 \\ -\frac{\alpha}{\sqrt{x}} + \alpha + 1, & x > 1, \end{cases} \tag{24}$$

where $\alpha$ is the thresholding coefficient. This approach ensures effective color correction and dynamic range handling during HDR-NSFF training.

We incorporate a smoothness loss to enforce that CRF varies smoothly in a physically plausible manner Debevec et al. (2023). We penalize the second-order derivative of the CRFs: It is defined as follows:

$$\mathcal{L}_{\text{smooth}} = \sum_{i=1}^{N} \sum_{e \in [0,1]} g_i''(e), \tag{25}$$

where $g''(e)$ denotes the second order derivative of CRFs *w.r.t.* its input domain.

In the absence of a known camera response function (CRF), the choice of tone-mapping module $\mathcal{T}(\cdot, \theta)$ determines the flexibility with which HDR radiance can be effectively recovered from LDR inputs. Moreover, to build consistent HDR representations in 3D space, the tone-mapping module must also act as a regularizer, preventing fluctuations in HDR results under multi-exposure conditions. This combination of flexibility and regularization largely influences the overall quality and stability of HDR field reconstruction.

### C.2 DINO-TRACKER

DINO-Tracker is a self-supervised framework designed to accurately track points over long sequences of video frames. Given an initial query point in an early frame of video, it estimates the trajectory of these points throughout subsequent frames. The method leverages pretrained deep features from the DINOv2-ViT (Oquab et al., 2023) model, which are refined by learning residual features via a small, trainable CNN module. DINO feature and residual feature are aggregated to find correspondence heatmap computed by cost volume. Lastly, additional CNN-refiner follows to further enhance matching.

Optimization is performed using several losses

- **Flow Loss ($L_{\text{flow}}$)**: Ensures predicted trajectories align closely with short-term optical flow correspondences.
- **DINO Best-Buddies Loss ($L_{\text{dino-bb}}$)**: Contrastively aligns refined features based on semantic matches from original DINO embeddings.
- **Refined Best-Buddies Loss ($L_{\text{rfn-bb}}$)**: Similar to DINO best-buddies loss but applied to newly detected reliable matches among refined features.
- **Cycle-Consistency Loss ($L_{\text{rfn-cc}}$)**: Encourages consistency in predicted trajectories, penalizing trajectories that fail a cycle-consistency criterion.
- **Prior Preservation Loss ($L_{\text{prior}}$)**: Regularizes the refined features to remain close in norm and direction to original DINO features, ensuring semantic coherence is preserved.

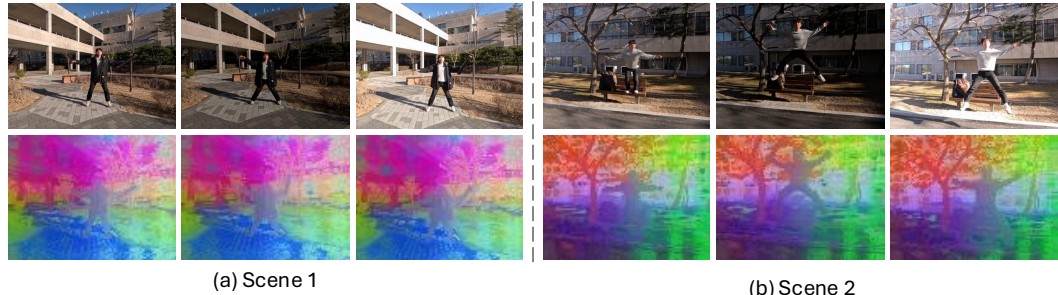

(a) Scene 1     (b) Scene 2

Figure S2: **DINOv2 feature visualization under varying exposures.** Despite large changes in brightness, DINOv2 embeddings remain consistent, showing robust clustering across different exposure levels.

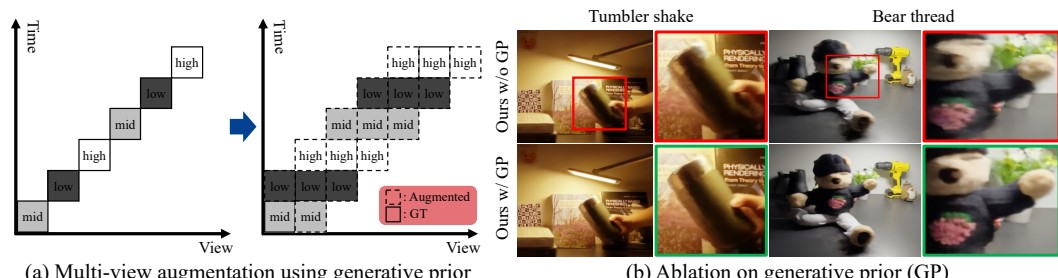

(a) Multi-view augmentation using generative prior     (b) Ablation on generative prior (GP)

Figure S3: **Effect of generative prior (GP).** (a) GP provides GP provides additional plausible views from unseen views, (b) leading to more consistent and sharper reconstructions.

In contrast to the original DINO-Tracker, our proposed approach introduces a novel utilization of this framework explicitly aimed at enhancing the robustness and accuracy of 2D dense correspondence estimation. Specifically, we propose deriving dense matching from consecutive frames using the trained DINO-Tracker model itself. Leveraging the semantic matching capability inherent to DINO features, our method provides robust optical flow estimates even in challenging conditions such as alternating-exposure video settings, where conventional texture-based methods typically degrade due to information loss. Figure S2 shows that DINOv2 features is robust to exposure variance.

### C.3 GENERATIVE PRIOR FOR RECOVERING SATURATED INFORMATION

In HDR-NSFF, we additionally employ generative prior (Wu et al., 2025) as a regularizer to stabilize training under severe exposure inconsistencies. Generative prior provides a diffusion-based enhancement prior that guides the radiance field toward semantically consistent reconstructions when input frames suffer from brightness fluctuations or missing details. Concretely, we periodically generate pseudo-observations by enhancing intermediate renderings with the Difix prior and incorporate them into the optimization loop. This regularization not only improves geometric and radiometric stability but also enforces stronger multi-view consistency in dynamic scenes, where exposure variations and motion often break correspondences across views. As a result, HDR-NSFF achieves more coherent reconstructions that generalize better to unseen exposures and viewpoints.

**Generative prior regularization**. To mitigate the sparse-view limitation of monocular input, we adopt enhanced views generated via a generative prior (Wu et al., 2025). For these views, we apply a patch-wise perceptual loss to encourage realistic and view-consistent appearance:

$$\mathcal{L}_{\text{gen}} = \sum_{p \in \mathcal{P}} \|\phi(\hat{C}_p) - \phi(C_p^{\text{gen}})\|_1, \tag{26}$$

where $\phi$ denotes a perceptual feature extractor, and $p$ indexes sampled patches. Since generative priors may introduce hallucinations, we carefully balance their contribution by (i) delaying their use until a stable stage of training ( 200K iterations), and (ii) training with enhanced views at a low probability (10%) per iteration.

| | Big jump | | | | | | Side walk | | | | | |
|---|---|---|---|---|---|---|---|---|---|---|---|---|
| Method | Full | | | Dynamic only | | | Full | | | Dynamic only | | |
| | PSNR | SSIM | LPIPS | PSNR | SSIM | LPIPS | PSNR | SSIM | LPIPS | PSNR | SSIM | LPIPS |
| NSFF | 16.23 | 0.6268 | 0.194 | 13.75 | 0.4476 | 0.2755 | 16.62 | 0.6346 | 0.1874 | 14.1 | 0.5524 | 0.2139 |
| 4DGS | 20.02 | 0.7283 | 0.1751 | 14.21 | 0.3724 | 0.319 | 18.46 | 0.702 | 0.1724 | 13.56 | 0.3927 | 0.254 |
| MotionGS | 11.4 | 0.1354 | 0.4549 | 9.42 | 0.0862 | 0.5628 | 13.58 | 0.2387 | 0.3692 | 8.79 | 0.1025 | 0.513 |
| NeRF-WT | 27.09 | 0.9051 | 0.0738 | 16.31 | 0.5023 | 0.2301 | 25.29 | 0.9061 | 0.0641 | 13.23 | 0.4243 | 0.2143 |
| HDR-Hexplane | 21.51 | 0.6235 | 0.2117 | 18.04 | 0.5653 | 0.2074 | 19.12 | 0.4931 | 0.2392 | 17.01 | 0.6214 | 0.1664 |
| Ours | 30.03 | 0.9239 | 0.0596 | 21.72 | 0.7494 | 0.1058 | 29.91 | 0.9263 | 0.0515 | 20.39 | 0.7132 | 0.1041 |

| | Jumping jack | | | | | | Pointing walk | | | | | |
|---|---|---|---|---|---|---|---|---|---|---|---|---|
| Method | Full | | | Dynamic only | | | Full | | | Dynamic only | | |
| | PSNR | SSIM | LPIPS | PSNR | SSIM | LPIPS | PSNR | SSIM | LPIPS | PSNR | SSIM | LPIPS |
| NSFF | 17.44 | 0.7543 | 0.1251 | 18.19 | 0.5493 | 0.1274 | 17.16 | 0.7534 | 0.1206 | 14.07 | 0.6198 | 0.1745 |
| 4DGS | 19.58 | 0.7486 | 0.1239 | 19.15 | 0.5733 | 0.1516 | 19.17 | 0.7373 | 0.1319 | 13.52 | 0.3664 | 0.2084 |
| MotionGS | 13.82 | 0.2474 | 0.3217 | 11.51 | 0.1244 | 0.4059 | 13.72 | 0.2472 | 0.3278 | 10.46 | 0.089 | 0.4774 |
| NeRF-WT | 30.93 | 0.9364 | 0.0384 | 19.94 | 0.6028 | 0.1384 | 27.36 | 0.9243 | 0.0488 | 15.26 | 0.5363 | 0.2018 |
| HDR-Hexplane | 17.24 | 0.4671 | 0.2112 | 19.89 | 0.5946 | 0.1466 | 16.97 | 0.4398 | 0.2111 | 16.67 | 0.6537 | 0.1674 |
| Ours | 31.83 | 0.9457 | 0.0353 | 23.66 | 0.7871 | 0.0721 | 29.72 | 0.9338 | 0.0433 | 20.05 | 0.7036 | 0.1046 |

| | Tube toss | | | | | | Bear thread | | | | | |
|---|---|---|---|---|---|---|---|---|---|---|---|---|
| Method | Full | | | Dynamic only | | | Full | | | Dynamic only | | |
| | PSNR | SSIM | LPIPS | PSNR | SSIM | LPIPS | PSNR | SSIM | LPIPS | PSNR | SSIM | LPIPS |
| NSFF | 17.75 | 0.7547 | 0.1237 | 16.94 | 0.7071 | 0.0913 | 15.45 | 0.566 | 0.4656 | 14.89 | 0.3535 | 0.5652 |
| 4DGS | 19.53 | 0.7429 | 0.127 | 17.72 | 0.6961 | 0.087 | 18.20 | 0.7811 | 0.2736 | 12.21 | 0.2426 | 0.5023 |
| MotionGS | 13.73 | 0.2469 | 0.3227 | 9.85 | 0.1433 | 0.3943 | 12.90 | 0.4995 | 0.494 | 9.83 | 0.1518 | 0.6414 |
| NeRF-WT | 31.63 | 0.9474 | 0.0315 | 19.44 | 0.8194 | 0.0704 | 22.13 | 0.8607 | 0.1618 | 13.55 | 0.2925 | 0.4098 |
| HDR-Hexplane | 17.13 | 0.4732 | 0.2211 | 16.43 | 0.6485 | 0.1096 | 22.08 | 0.7903 | 0.2597 | 18.09 | 0.5367 | 0.3357 |
| Ours | 32.08 | 0.9482 | 0.0348 | 24.78 | 0.9105 | 0.0366 | 30.19 | 0.9224 | 0.1337 | 23.93 | 0.7568 | 0.2448 |

| | Tumbler | | | | | | Dog | | | | | |
|---|---|---|---|---|---|---|---|---|---|---|---|---|
| Method | Full | | | Dynamic only | | | Full | | | Dynamic only | | |
| | PSNR | SSIM | LPIPS | PSNR | SSIM | LPIPS | PSNR | SSIM | LPIPS | PSNR | SSIM | LPIPS |
| NSFF | 17.8 | 0.5645 | 0.2897 | 14.89 | 0.4618 | 0.3354 | 17.87 | 0.5403 | 0.2884 | 14.96 | 0.5193 | 0.3316 |
| 4DGS | 22.05 | 0.8171 | 0.1753 | 17.54 | 0.6221 | 0.2124 | 27.15 | 0.9164 | 0.1069 | 19.42 | 0.7279 | 0.1661 |
| MotionGS | 15.41 | 0.5143 | 0.4434 | 12.16 | 0.3455 | 0.5881 | 15.26 | 0.4963 | 0.4309 | 11.73 | 0.3149 | 0.5667 |
| NeRF-WT | 31.97 | 0.9456 | 0.0625 | 20.48 | 0.7477 | 0.1733 | 31.7 | 0.9466 | 0.0719 | 20.15 | 0.7525 | 0.143 |
| HDR-Hexplane | 24.98 | 0.825 | 0.1828 | 22.24 | 0.7556 | 0.1909 | 25.02 | 0.8445 | 0.1905 | 20.46 | 0.7165 | 0.2318 |
| Ours | 34.92 | 0.9428 | 0.0727 | 27.85 | 0.8863 | 0.0927 | 33.14 | 0.944 | 0.0841 | 24.85 | 0.8422 | 0.1441 |

| | Fire extinguisher | | | | | | Laptop | | | | | |
|---|---|---|---|---|---|---|---|---|---|---|---|---|
| Method | Full | | | Dynamic only | | | Full | | | Dynamic only | | |
| | PSNR | SSIM | LPIPS | PSNR | SSIM | LPIPS | PSNR | SSIM | LPIPS | PSNR | SSIM | LPIPS |
| NSFF | 21.04 | 0.8215 | 0.1300 | 26.5 | 0.6232 | 0.1742 | 23.17 | 0.8437 | 0.102 | 27.41 | 0.7219 | 0.1107 |
| 4DGS | 22.73 | 0.8568 | 0.1324 | 23.94 | 0.6189 | 0.2659 | 23.86 | 0.8818 | 0.1023 | 28.34 | 0.8203 | 0.0957 |
| MotionGS | 18.34 | 0.6445 | 0.2728 | 20.56 | 0.4259 | 0.4292 | 18.64 | 0.6568 | 0.2378 | 20.8 | 0.4533 | 0.2908 |
| NeRF-WT | 32.78 | 0.9506 | 0.0496 | 24.39 | 0.6706 | 0.2079 | 37.4 | 0.9801 | 0.0197 | 29.26 | 0.9034 | 0.048 |
| HDR-Hexplane | 23.57 | 0.8876 | 0.0797 | 28.78 | 0.7612 | 0.080 | 23.51 | 0.8993 | 0.0721 | 29.9 | 0.8375 | 0.0611 |
| Ours | 36.82 | 0.9686 | 0.0298 | 32.04 | 0.8494 | 0.0777 | 37.28 | 0.9759 | 0.0231 | 33.32 | 0.9141 | 0.0407 |

| | Bag | | | | | | Ball | | | | | |
|---|---|---|---|---|---|---|---|---|---|---|---|---|
| Method | Full | | | Dynamic only | | | Full | | | Dynamic only | | |
| | PSNR | SSIM | LPIPS | PSNR | SSIM | LPIPS | PSNR | SSIM | LPIPS | PSNR | SSIM | LPIPS |
| NSFF | 18.33 | 0.6867 | 0.2029 | 19.32 | 0.5594 | 0.1516 | 17.38 | 0.6033 | 0.2433 | 16.08 | 0.4525 | 0.2439 |
| 4DGS | 19.54 | 0.7723 | 0.1744 | 18.13 | 0.6577 | 0.1706 | 21.01 | 0.8016 | 0.1534 | 16.17 | 0.5384 | 0.2431 |
| MotionGS | 14.59 | 0.4542 | 0.3284 | 12.43 | 0.3155 | 0.3460 | 13.95 | 0.3903 | 0.3366 | 10.47 | 0.2114 | 0.4195 |
| NeRF-WT | 29.9 | 0.9518 | 0.0488 | 22.34 | 0.7912 | 0.09 | 28.26 | 0.9453 | 0.0461 | 16.7 | 0.5584 | 0.1974 |
| HDR-Hexplane | 18.84 | 0.6643 | 0.2116 | 19.82 | 0.6863 | 0.1631 | 18.38 | 0.6246 | 0.2101 | 19.3 | 0.5772 | 0.1992 |
| Ours | 32.53 | 0.9532 | 0.0495 | 27.13 | 0.8937 | 0.0603 | 33.11 | 0.9482 | 0.0473 | 26.22 | 0.8431 | 0.0827 |

Table S2: **Quantitative results of novel view synthesis on GoPro dataset.** The green and yellow colors stand for the best and the second best, respectively.

### C.4 OBJECTIVE FUNCTION

Finally, our HDR-NSFF is end-to-end optimized using the following loss:

$$\mathcal{L} = \mathcal{L}_{cb} + \mathcal{L}_{photo} + \beta_{data}\mathcal{L}_{data} + \beta_{reg}\mathcal{L}_{reg} + \beta_{smooth}\mathcal{L}_{smooth} + \beta_{enh}\mathcal{L}_{gen}, \qquad (27)$$

where the $\beta$ coefficients weight each term. Additional regularization terms, $\mathcal{L}_{reg}$ leveraging scene flow priors.

## D ADDITIONAL EXPERIMENT RESULTS

We provide additional experimental results that could not be included in the main manuscript, due to page limit. Specifically, Tables S2, S3, & S4 present per-scene quantitative results for each experiment. Figures S8-S19 illustrates qualitative outcomes for additional real datasets not shown in

| Method | Big jump – Full | | | Big jump – Dynamic only | | | Side walk – Full | | | Side walk – Dynamic only | | |
|---|---|---|---|---|---|---|---|---|---|---|---|---|
| | PSNR | SSIM | LPIPS | PSNR | SSIM | LPIPS | PSNR | SSIM | LPIPS | PSNR | SSIM | LPIPS |
| NSFF | 17.09 | 0.6856 | 0.1858 | 15.05 | 0.4765 | 0.3175 | 17.48 | 0.7095 | 0.1621 | 15.49 | 0.609 | 0.2292 |
| HDR-Hexplane | 19.01 | 0.5076 | 0.2301 | 15.13 | 0.4166 | 0.2604 | 19.08 | 0.4914 | 0.2389 | 16.33 | 0.5778 | 0.1699 |
| Ours | 30.13 | 0.924 | 0.0662 | 22.09 | 0.756 | 0.1515 | 30.18 | 0.927 | 0.0564 | 21.03 | 0.7304 | 0.1413 |

| Method | Jumping jack – Full | | | Jumping jack – Dynamic only | | | Pointing walk – Full | | | Pointing walk – Dynamic only | | |
|---|---|---|---|---|---|---|---|---|---|---|---|---|
| | PSNR | SSIM | LPIPS | PSNR | SSIM | LPIPS | PSNR | SSIM | LPIPS | PSNR | SSIM | LPIPS |
| NSFF | 18.26 | 0.7873 | 0.1212 | 19.82 | 0.62 | 0.1448 | 18.21 | 0.7752 | 0.1286 | 15.56 | 0.6301 | 0.248 |
| HDR-Hexplane | 17.22 | 0.4626 | 0.2134 | 19.56 | 0.5833 | 0.1471 | 16.98 | 0.4383 | 0.2107 | 16.4 | 0.6171 | 0.1702 |
| Ours | 32.03 | 0.9457 | 0.0395 | 24.2 | 0.7954 | 0.0995 | 30.05 | 0.9346 | 0.0473 | 20.74 | 0.7194 | 0.1287 |

| Method | Tube toss – Full | | | Tube toss – Dynamic only | | | Bear thread – Full | | | Bear thread – Dynamic only | | |
|---|---|---|---|---|---|---|---|---|---|---|---|---|
| | PSNR | SSIM | LPIPS | PSNR | SSIM | LPIPS | PSNR | SSIM | LPIPS | PSNR | SSIM | LPIPS |
| NSFF | 18.5 | 0.7968 | 0.1095 | 18.4 | 0.7628 | 0.0967 | 16.91 | 0.5648 | 0.4884 | 15.96 | 0.3872 | 0.5795 |
| HDR-Hexplane | 17.15 | 0.4737 | 0.2204 | 16.35 | 0.6457 | 0.1096 | 21.81 | 0.785 | 0.2604 | 17.35 | 0.493 | 0.3452 |
| Ours | 32.27 | 0.9482 | 0.0378 | 25.19 | 0.9125 | 0.0498 | 30.4 | 0.9235 | 0.1437 | 24.29 | 0.7626 | 0.2869 |

| Method | Dog – Full | | | Dog – Dynamic only | | | Tumbler – Full | | | Tumbler – Dynamic only | | |
|---|---|---|---|---|---|---|---|---|---|---|---|---|
| | PSNR | SSIM | LPIPS | PSNR | SSIM | LPIPS | PSNR | SSIM | LPIPS | PSNR | SSIM | LPIPS |
| NSFF | 18.42 | 0.5972 | 0.3588 | 15.28 | 0.4708 | 0.3672 | 18.84 | 0.6352 | 0.2457 | 16.25 | 0.4823 | 0.3693 |
| HDR-Hexplane | 25.02 | 0.8438 | 0.1881 | 17.38 | 0.4989 | 0.2992 | 24.93 | 0.8247 | 0.1837 | 20.42 | 0.6685 | 0.2142 |
| Ours | 33.45 | 0.9452 | 0.0875 | 21.16 | 0.6793 | 0.2781 | 35.14 | 0.944 | 0.0752 | 26.12 | 0.8476 | 0.1415 |

| Method | Fire extinguisher – Full | | | Fire extinguisher – Dynamic only | | | Laptop – Full | | | Laptop – Dynamic only | | |
|---|---|---|---|---|---|---|---|---|---|---|---|---|
| | PSNR | SSIM | LPIPS | PSNR | SSIM | LPIPS | PSNR | SSIM | LPIPS | PSNR | SSIM | LPIPS |
| NSFF | 22.94 | 0.8639 | 0.1094 | 27.67 | 0.6782 | 0.1842 | 24.19 | 0.8908 | 0.0760 | 28.87 | 0.7801 | 0.0971 |
| HDR-Hexplane | 23.56 | 0.8854 | 0.0804 | 27.84 | 0.7416 | 0.0847 | 23.54 | 0.8992 | 0.0719 | 29.67 | 0.8359 | 0.0610 |
| Ours | 37.00 | 0.9694 | 0.0331 | 32.46 | 0.8568 | 0.0984 | 37.41 | 0.9763 | 0.0237 | 33.72 | 0.9187 | 0.0454 |

| Method | Bag – Full | | | Bag – Dynamic only | | | Ball – Full | | | Ball – Dynamic only | | |
|---|---|---|---|---|---|---|---|---|---|---|---|---|
| | PSNR | SSIM | LPIPS | PSNR | SSIM | LPIPS | PSNR | SSIM | LPIPS | PSNR | SSIM | LPIPS |
| NSFF | 19.33 | 0.7352 | 0.1815 | 20.51 | 0.6336 | 0.1439 | 17.95 | 0.6685 | 0.2042 | 17.25 | 0.5172 | 0.2595 |
| HDR-Hexplane | 18.85 | 0.6641 | 0.2116 | 19.76 | 0.6818 | 0.1627 | 18.36 | 0.6243 | 0.2104 | 18.91 | 0.5684 | 0.2014 |
| Ours | 32.26 | 0.9525 | 0.0525 | 26.91 | 0.8873 | 0.0748 | 32.64 | 0.9468 | 0.0499 | 25.25 | 0.8185 | 0.1105 |

Table S3: **Quantitative results of novel time synthesis on real data.** The green and yellow colors stand for the  best  and the  second best , respectively.

| Methods | synthetic dataset-Full | | | | Methods | synthetic dataset-Dynamic | | | |
|---|---|---|---|---|---|---|---|---|---|
| | Lego | Mutant | Standup | Jumping Jack | | Lego | Mutant | Standup | Jumping Jack |
| | PSNR | | | | | PSNR | | | |
| NSFF | 15.45 | 16.97 | 13.47 | 15.53 | NSFF | 15.94 | 18.43 | 10.25 | 13.74 |
| NeRF-WT | 29.55 | 33.06 | 32.55 | 29.25 | NeRF-WT | 22.32 | 27.58 | 19.77 | 16.33 |
| HDR-Hexplane | 28.58 | 30.88 | 30.83 | 29.50 | HDR-Hexplane | 24.61 | 29.71 | 21.59 | 19.57 |
| Ours | 34.64 | 36.13 | 35.80 | 33.72 | Ours | 28.77 | 31.80 | 24.98 | 23.21 |
| | SSIM | | | | | SSIM | | | |
| NSFF | 0.6472 | 0.6348 | 0.4958 | 0.6551 | NSFF | 0.6145 | 0.5152 | 0.1601 | 0.5795 |
| NeRF-WT | 0.9595 | 0.9114 | 0.9556 | 0.9200 | NeRF-WT | 0.8517 | 0.8289 | 0.7741 | 0.5412 |
| HDR-Hexplane | 0.9443 | 0.8526 | 0.9112 | 0.9137 | HDR-Hexplane | 0.8626 | 0.8443 | 0.7665 | 0.7262 |
| Ours | 0.9670 | 0.9278 | 0.9564 | 0.9348 | Ours | 0.9062 | 0.9115 | 0.8816 | 0.8349 |
| | LPIPS | | | | | LPIPS | | | |
| NSFF | 0.1556 | 0.1243 | 0.2368 | 0.1364 | NSFF | 0.1528 | 0.1708 | 0.3097 | 0.1345 |
| NeRF-WT | 0.0171 | 0.0316 | 0.0224 | 0.0655 | NeRF-WT | 0.0592 | 0.0845 | 0.0988 | 0.1154 |
| HDR-Hexplane | 0.0257 | 0.0708 | 0.0603 | 0.0539 | HDR-Hexplane | 0.1217 | 0.0724 | 0.1547 | 0.0794 |
| Ours | 0.0147 | 0.0305 | 0.0249 | 0.1229 | Ours | 0.0426 | 0.0590 | 0.0749 | 0.0538 |

Table S4: **Quantitative results of novel view and time synthesis on synthetic dataset.** The green and yellow colors stand for the  best  and the  second best , respectively.

the main paper. Moreover, supplementary videos include more HDR, LDR, and novel view rendering results. Please refer supplementary video for further visualization results.

## D.1 ABLATION STUDY

We analyze the impact of our proposed semantic-based optical flow on the novel view synthesis task using 8 real dataset samples. We compare two variants of our method: (1) Ours (w/ RAFT), in which the RAFT optical flow is used without modification, and (2) Ours (w/ RAFT Finetuned), where RAFT is fine-tuned on synthetic multi-exposure data. Note that, as shown in Figure 4, the original RAFT model was not trained on multi-exposed images, resulting in high errors when applied directly in our

| Methods | Full | | | Dynamic only | | |
|---------|------|------|------|------|------|------|
| | PSNR↑ | SSIM↑ | LPIPS↓ | PSNR↑ | SSIM↑ | LPIPS↓ |
| Ours w/ RAFT (Teed and Deng, 2020) | 30.42 | 0.9269 | 0.0246 | 21.38 | 0.7369 | 0.0675 |
| Ours w/ Finetuned | 30.68 | 0.9234 | 0.0253 | 21.51 | 0.7377 | 0.0689 |
| Ours w/ Dino-Tracker (Tumanyan et al., 2024) | 31.01 | 0.9301 | 0.0233 | 22.55 | 0.7714 | 0.0697 |

Table S5: **Ablation study of flow model.** To compare the effect of flow regularization, we compare NVS performance of our approach against the baseline optical flow model (RAFT Teed and Deng (2020)) and a stronger baseline fine-tuned RAFT on a multi-exposure adaptation of the FlyingThings3D dataset.

| Methods | Full | | | Dynamic only | | |
|---------|------|------|------|------|------|------|
| | PSNR↑ | SSIM↑ | LPIPS↓ | PSNR↑ | SSIM↑ | LPIPS↓ |
| MoSca w/ TM | 26.67 | 0.920 | 0.060 | 15.96 | 0.463 | 0.064 |
| MoSca w/ TM + DT | **29.35** | **0.937** | **0.040** | **21.22** | **0.751** | **0.064** |

Table S6: **Quantitative results of MoSca with our proposed modules on GoPro outdoor scenes for HDR novel view synthesis.** MoSca with Tone-mapping (TM) aligns exposure-varied inputs into a consistent HDR radiance but still suffers from unreliable motion due to CoTracker's exposure sensitivity. Incorporating DINO-Tracker (DT) provides exposure-robust semantic flow, significantly improving both HDR reconstruction and dynamic motion stability.

setting. By fine-tuning it on synthetic data, the performance is improved. As shown in Table S5, our proposed method achieves the best results.

# E    EXTENSION TO 3DGS-BASED DYNAMIC SCENE RECONSTRUCTION

While our method is developed on top of NSFF, its core components—tone-mapping and exposure-robust semantic flow via DINO-Tracker—are not tied to the NSFF framework. In this section, we demonstrate that our approach is broadly applicable and can be seamlessly integrated into 3D Gaussian Splatting (3DGS)–based dynamic reconstruction pipelines.

**Applicability beyond NSFF**. The primary objective of our design is to make dynamic 4D reconstruction robust to the exposure variance inherent in alternating-exposure monocular videos. To verify that the proposed components are method-agnostic, we extend our pipeline to a representative 3DGS-based method, MoSca (Lei et al., 2025). For this purpose, we integrate our modules into MoSca without modifying its core architecture. First, our learnable tone-mapping (TM) module transforms LDR frames acquired under varying exposures into a unified HDR radiance space, providing exposure-invariant appearance supervision throughout the optimization. In addition, we replace MoSca's original motion estimation, CoTracker (Karaev et al., 2024) with semantic flow obtained from DINO-Tracker(DT), which offers robust correspondence cues under severe illumination and exposure variations. With these modifications, we assess whether the proposed components can still enhance HDR dynamic reconstruction in a framework that depends on explicit Gaussian tracking rather than continuous scene-flow modeling.

**Experimental setup**. We run all experiments on the proposed GoPro outdoor dataset. For this ablation, we compare two variants of MoSca augmented with our components: MoSca + TM, which incorporates only our tone-mapping (TM) module, and MoSca + TM + DT, our full configuration equipped with both exposure-aware appearance normalization and robust semantic flow (DINO-Tracker, DT). We note that the original MoSca configuration with CoTracker-based motion cues consistently failed to converge under alternating-exposure inputs. Thus, we exclude it from comparison. We evaluate novel view synthesis (NVS) results and report quantitative metrics in addition to qualitative visualizations.

**Quantitative results**. Table S6 shows that applying only the tone-mapping module improves the HDR appearance reconstruction, as it successfully aligns exposure-varied images into a common radiometric domain. However, this configuration struggles to recover consistent motion: the CoTracker-based correspondences often fail under extreme exposure variations, leading to incorrect dynamic geometry. When both TM and DINO-Tracker are applied, the model achieves the best performance across all

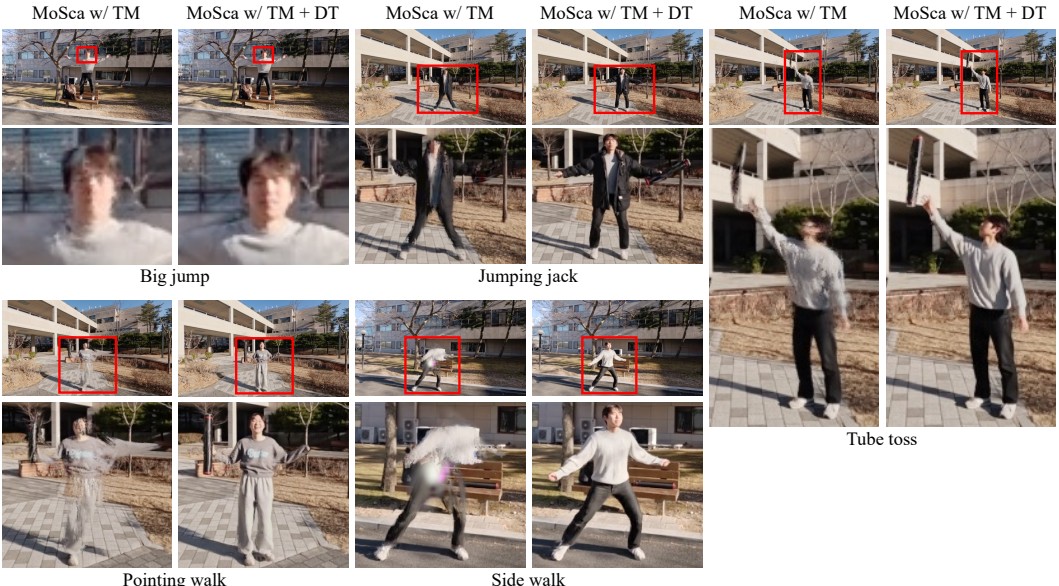

Figure S4: **Qualitative results of MoSca with our proposed modules on GoPro outdoor scenes for HDR novel view synthesis.** MoSca with Tone-mapping module (TM) produces HDR-like appearances but suffers from temporal inconsistency and distorted dynamic geometry due to exposure-sensitive motion estimation. In contrast, MoSca with TM and DINO-Tracker (DT) leverages exposure-robust semantic flow, yielding stable geometry and photometrically consistent HDR novel views under challenging exposure alternation.

metrics. The semantic flow provides exposure-robust motion cues, enabling the 3DGS optimization to recover temporally stable Gaussian trajectories and coherent HDR appearance over time.

**Qualitative results**. Figure S4 shows qualitative results of MoSca based our approach on our GoPro Outdoor scenes. While MoSca+TM produces HDR-like frames, the reconstructed geometry becomes inconsistent across time due to unreliable motion supervision. In contrast, MoSca+TM+DT produces stable and photometrically consistent HDR novel views, successfully handling complex exposure alternation and large dynamic motions.

These results demonstrate that the improvements brought by our approach—our tone-mapping module and exposure-robust semantic flow—are not specific to NSFF. The same components substantially enhance a 3DGS-based method, improving both radiance reconstruction and motion stability. This indicates that the proposed modules provide general utility for 4D HDR reconstruction from alternating-exposure monocular videos, regardless of the underlying 3D representation.

# F  2D-TO-4D HDR RECONSTURCTION

We compare our method against a two-stage baseline that first reconstructs an HDR video from alternating-exposure input frames using existing 2D HDR video methods—LAN-HDR (Chung and Cho, 2023), HDRFlow (Xu et al., 2024), and NECHDR (Cui et al., 2024)—and subsequently applies a dynamic 4D reconstruction framework. For each method, the predicted HDR frames are tone-mapped using a fixed $\mu$-law operator, which compresses HDR radiance values $E$ into the LDR domain through a logarithmic mapping:

$$M(E) = \frac{\log(1 + \mu E)}{\log(1 + \mu)}, \tag{28}$$

where $\mu = 500$ controls the compression strength. The tone-mapped frames are then provided as input to MoSca (Lei et al., 2025) for 4D reconstruction. For a fair comparison with the two-stage 2D-to-4D baseline, which uses MoSca for 4D reconstruction, we additionally report a MoSca-based variant of our pipeline (HDR-MoSca). This isolates the effect of the 2D HDR reconstruction stage from the choice of 4D representation.

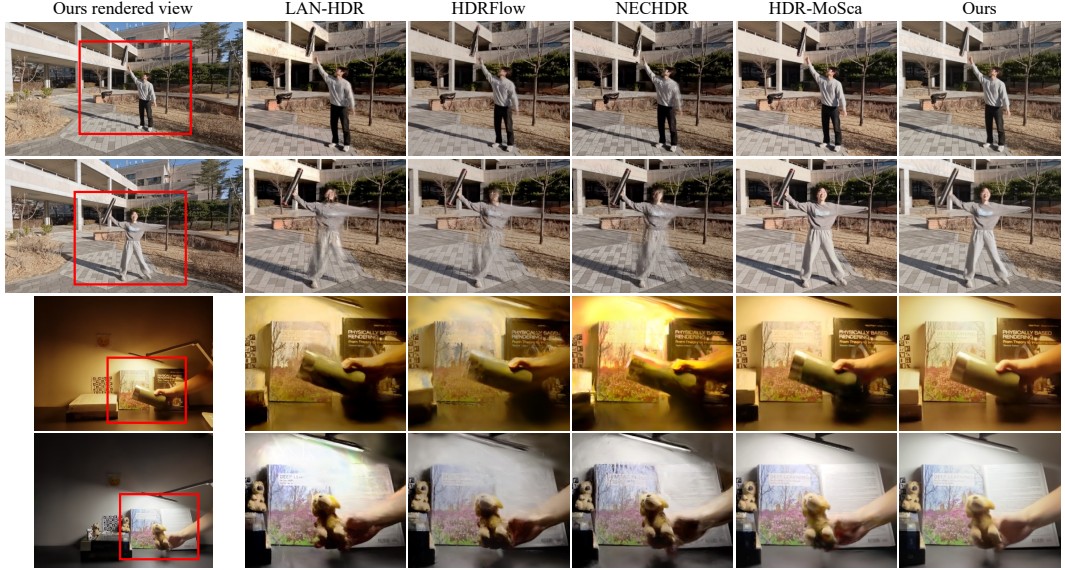

Figure S5: **Qualitative comparison with 2D-to-4D HDR reconstruction.** A two-stage baseline reconstructs HDR video using LAN-HDR (Chung and Cho, 2023), HDRFlow (Xu et al., 2024), or NECHDR (Cui et al., 2024), applies a fixed $\mu$-law tone mapping, and performs 4D reconstruction with MoSca (Lei et al., 2025). For reference, we also include a MoSca-based variant of our method (HDR-MoSca). While 2D-to-4D baselines fail to recover, our approach yields temporally coherent HDR radiance and stable geometry.

While these 2D HDR video methods produce visually plausible results under mild motion, they fundamentally operate within the 2D image plane and therefore inherit the limitations outlined in Figure 2. In scenarios with noticeable camera motion they struggle to reliably handle occlusions, complex dynamics, and the severe exposure inconsistency intrinsic to alternating-exposure videos.

When such inconsistent HDR frames are used for 4D reconstruction, the downstream model receives observations that are radiometrically unstable and geometrically incoherent, preventing reliable estimation of density, radiance, and dynamic motion. Since the second stage has no mechanism to correct errors originating from the 2D reconstruction stage, these radiometric inconsistencies propagate forward and degrade the overall quality of the 4D reconstruction. Figure S5 shows that the two-stage 2D-to-4D baseline often produces temporally inconsistent radiance fields and fails to reconstruct stable geometry under exposure variation.

In contrast, our method performs end-to-end dynamic HDR radiance reconstruction directly from alternating-exposure inputs, jointly reasoning about radiance, geometry, and motion within a unified 4D representation. This formulation leverages geometric and motion priors unavailable to 2D methods, enabling consistent tone reproduction, recovery of valid information in saturated regions, and significantly more stable reconstruction under challenging exposure variations.

## G   QUALITATIVE RESULTS ON HDR-GOPRO DATASET

Figure S6 and S7 show training view and ours reconstructed views on HDR-GoPro dataset. Figure S8-S19 show qualitative results of novel view synthesis on HDR-GoPro dataset.

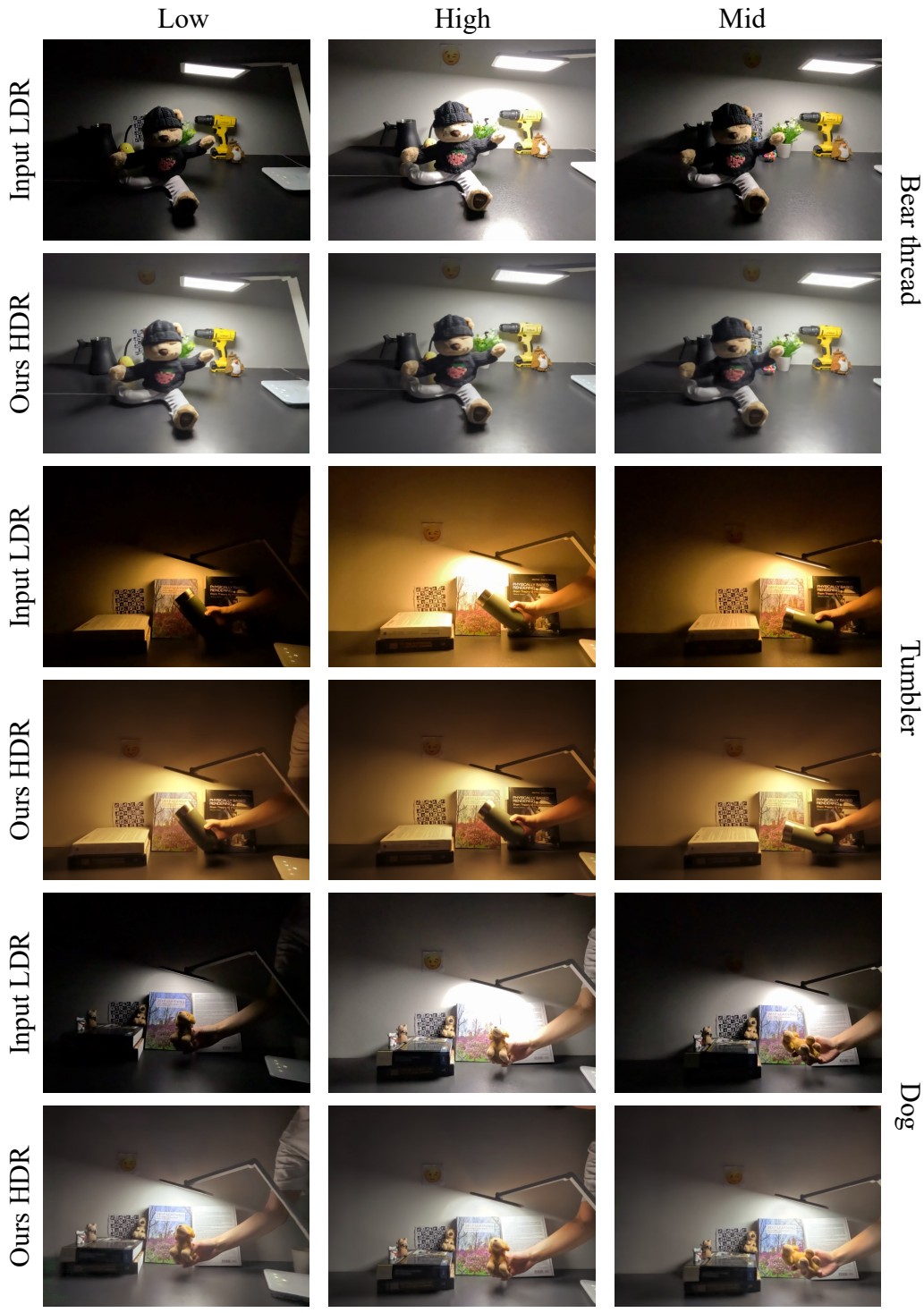

Figure S6: **Training view and ours reconstrcuted views on GoPro indoor dataset.** The odd-numbered rows show sample LDR frames from the input sequence, captured with alternating exposures (low, high, and mid). The even-numbered rows present our tone-mapped HDR reconstruction results for the corresponding input views. Under- and over-exposed regions are reliably recovered across all scenes. Even in areas where severe saturation is expected, such as the smile-emoji picture on the wall or the region directly beneath the light source, our method accurately reconstructs fine details. Even the mid-exposure frames contain locally saturated regions, yet these areas are consistently restored with high fidelity.

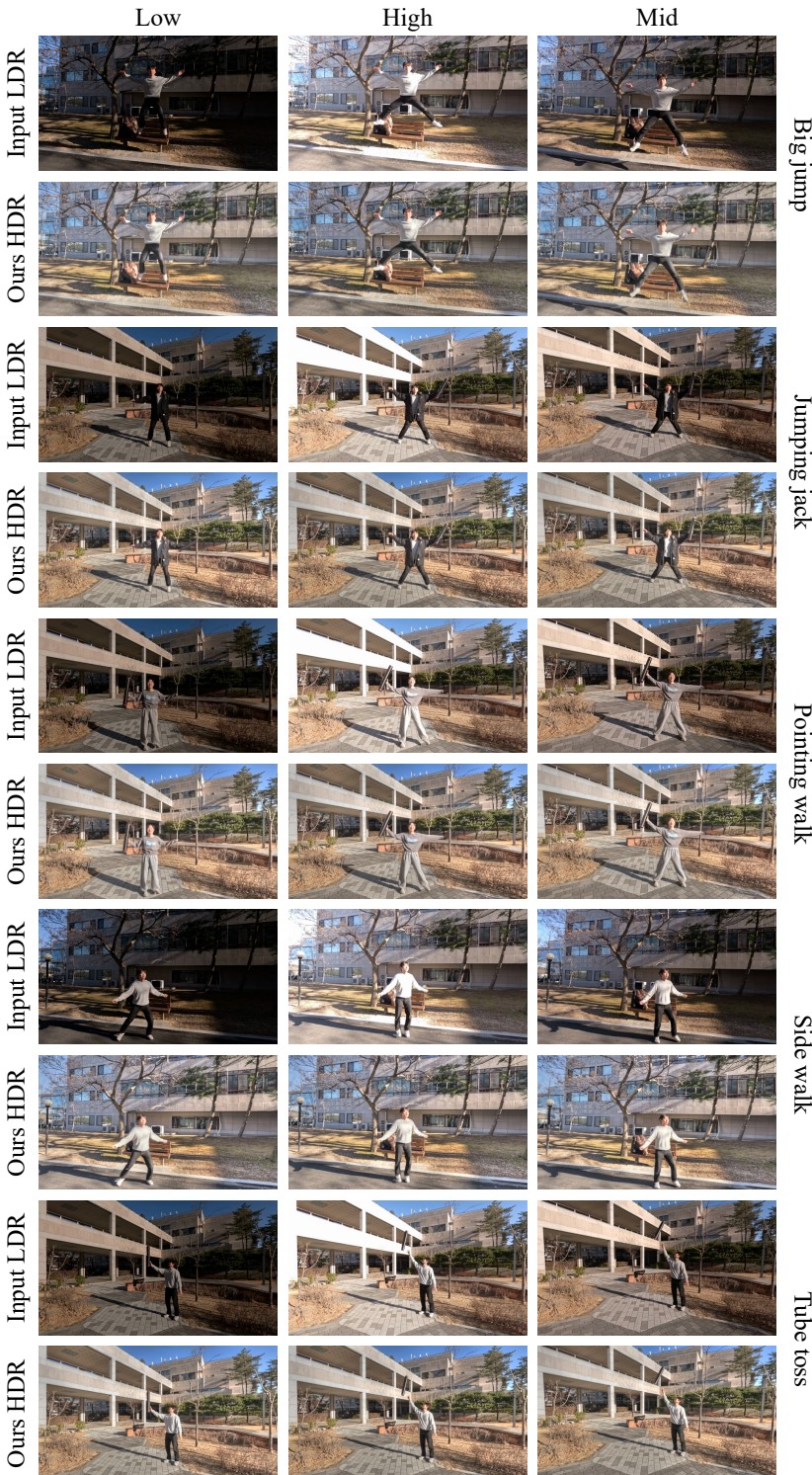

Figure S7: **Training view and ours reconstrcuted views on GoPro outdoor dataset.** The odd-numbered rows show sample LDR frames from the input sequence, captured with alternating exposures (low, high, and mid). The even-numbered rows present our tone-mapped HDR reconstruction results for the corresponding input views. Across the outdoor scenes, our method robustly reconstructs HDR radiance even under strong and diverse motion patterns. It performs reliably in scenarios involving large vertical motion in Big Jump, lateral motion in Pointing Walk, and fast object motion in Tube Toss. Despite these challenging dynamics, the model successfully restores both saturated regions caused by strong sunlight reflections on buildings and under-exposed regions cast in shadow, demonstrating consistent reconstruction quality across a wide range of exposure conditions.

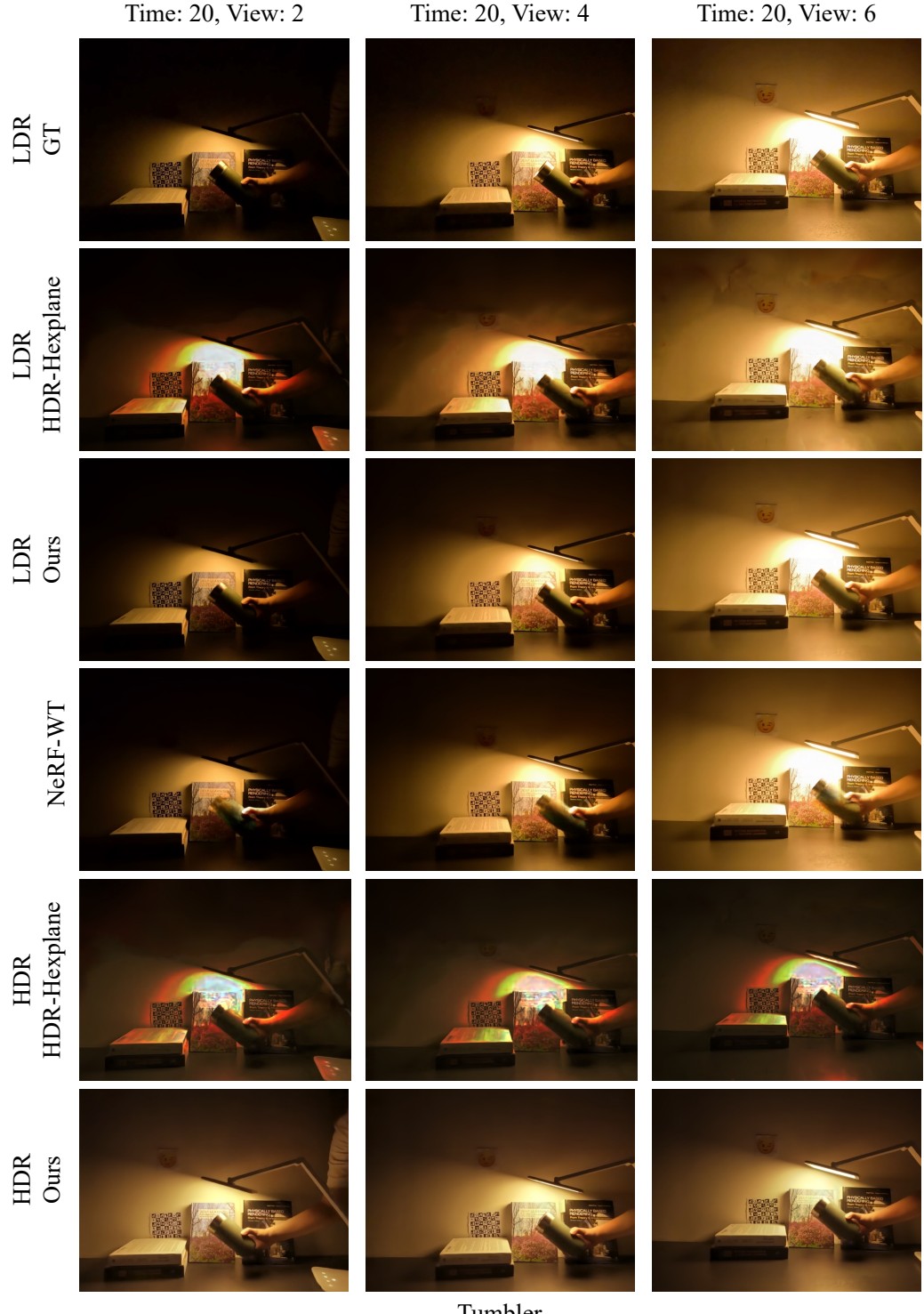

Figure S8: **Qualitative results of novel view synthesis on *Tumbler* data.**

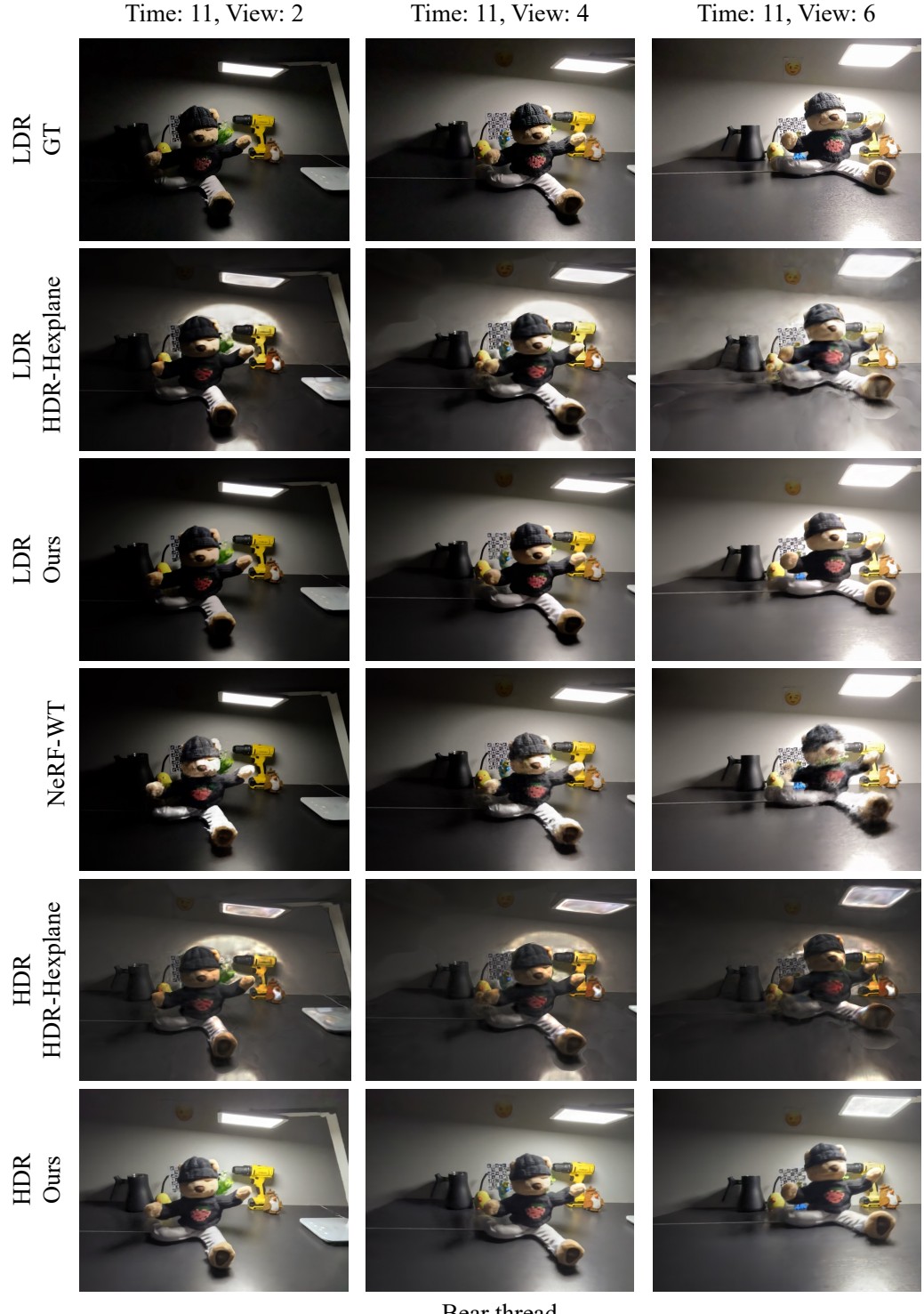

Bear thread

Figure S9: **Qualitative results of novel view synthesis on *Bear thread* data.**

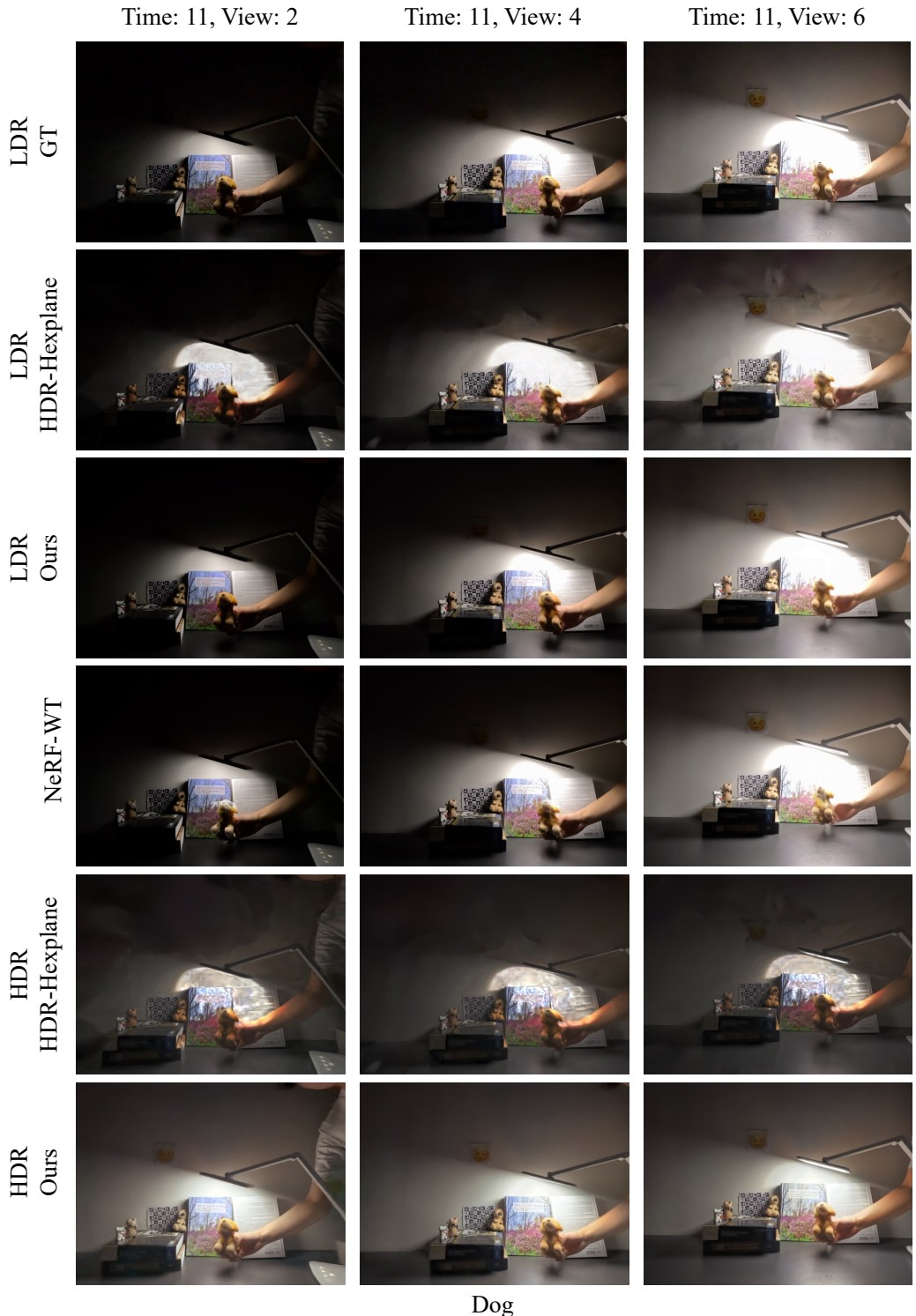

Figure S10: **Qualitative results of novel view synthesis on *Dog* data.**

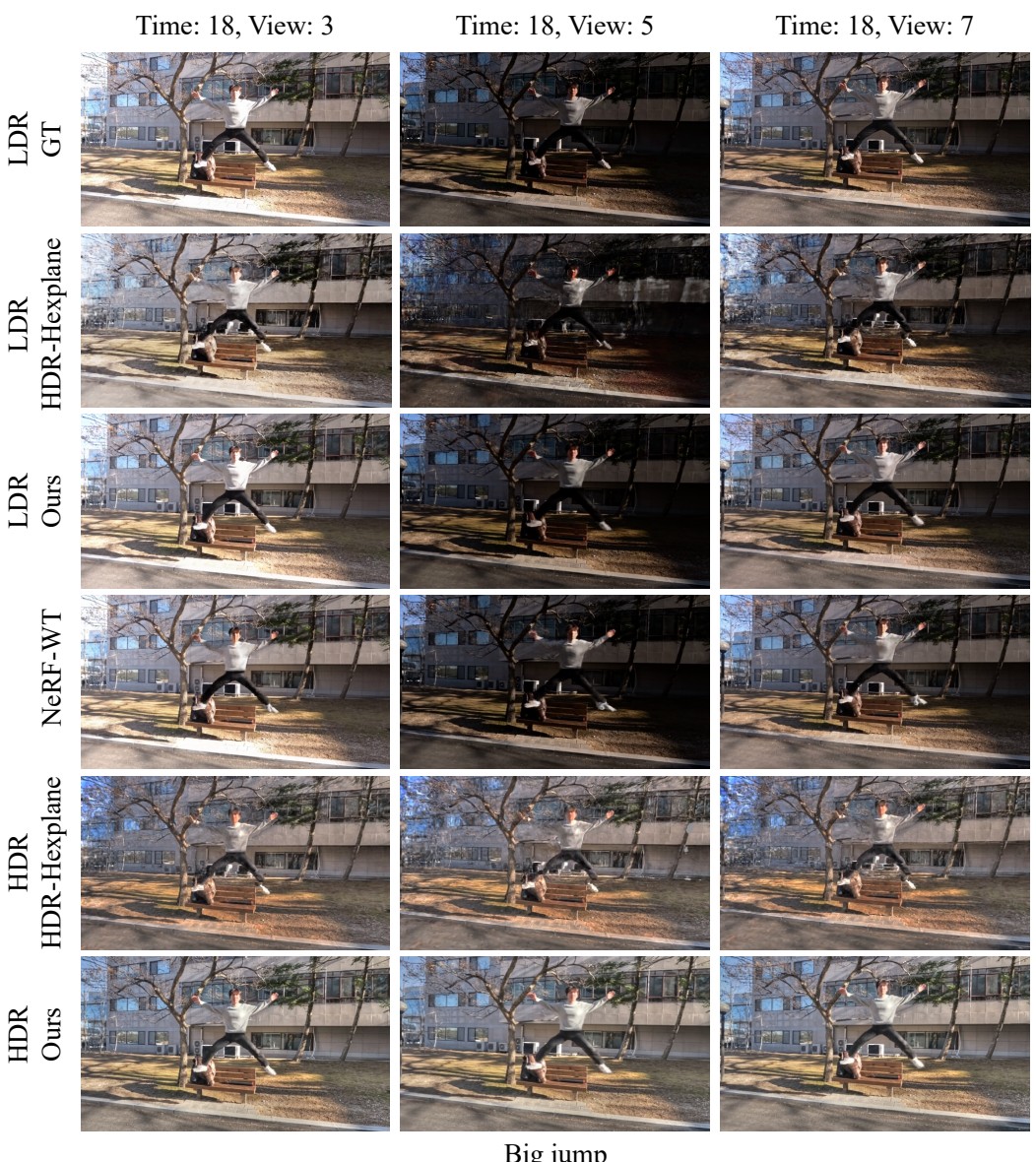

Figure S11: **Qualitative results of novel view synthesis on *Big jump* data.**

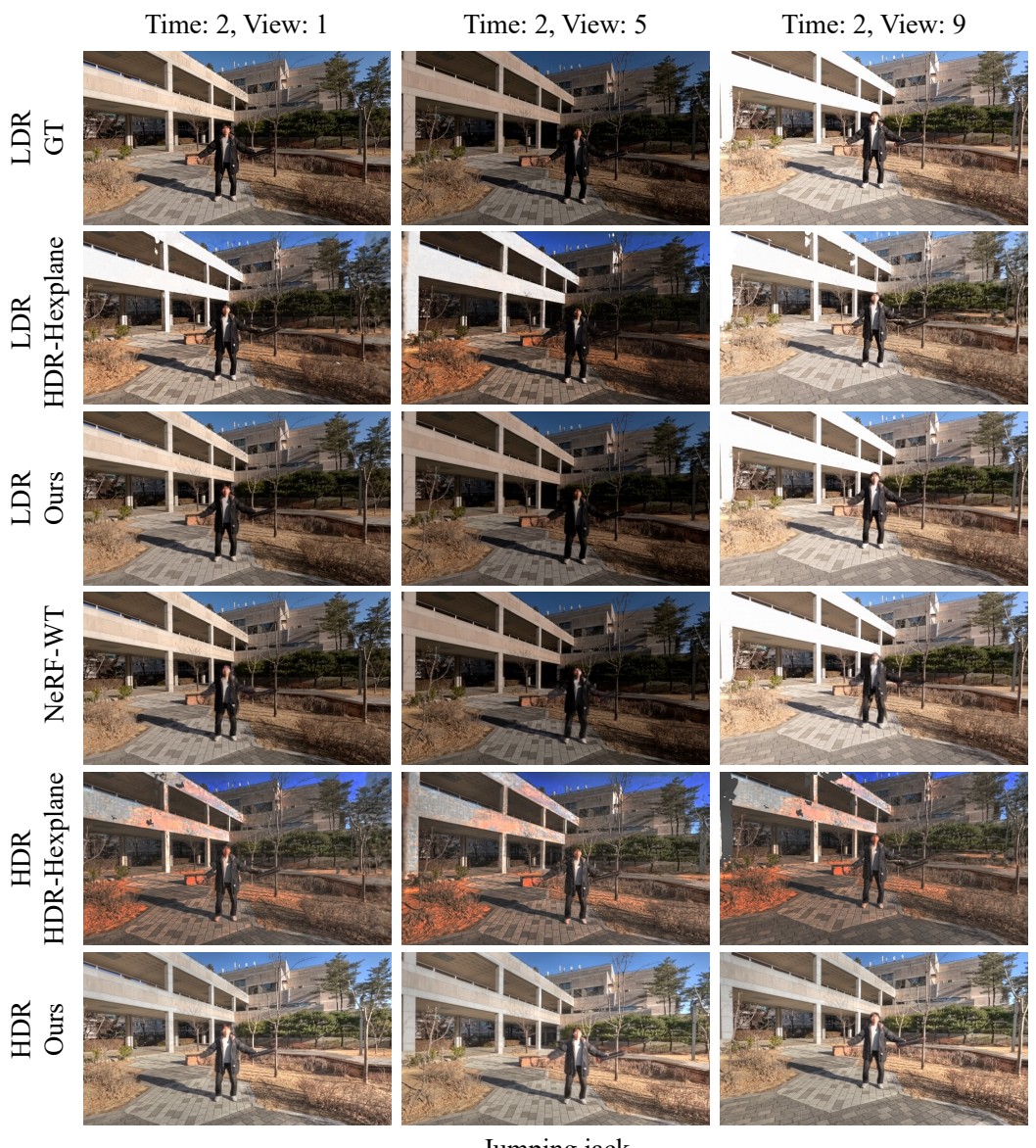

Figure S12: **Qualitative results of novel view synthesis on *Jumping jack* data.**

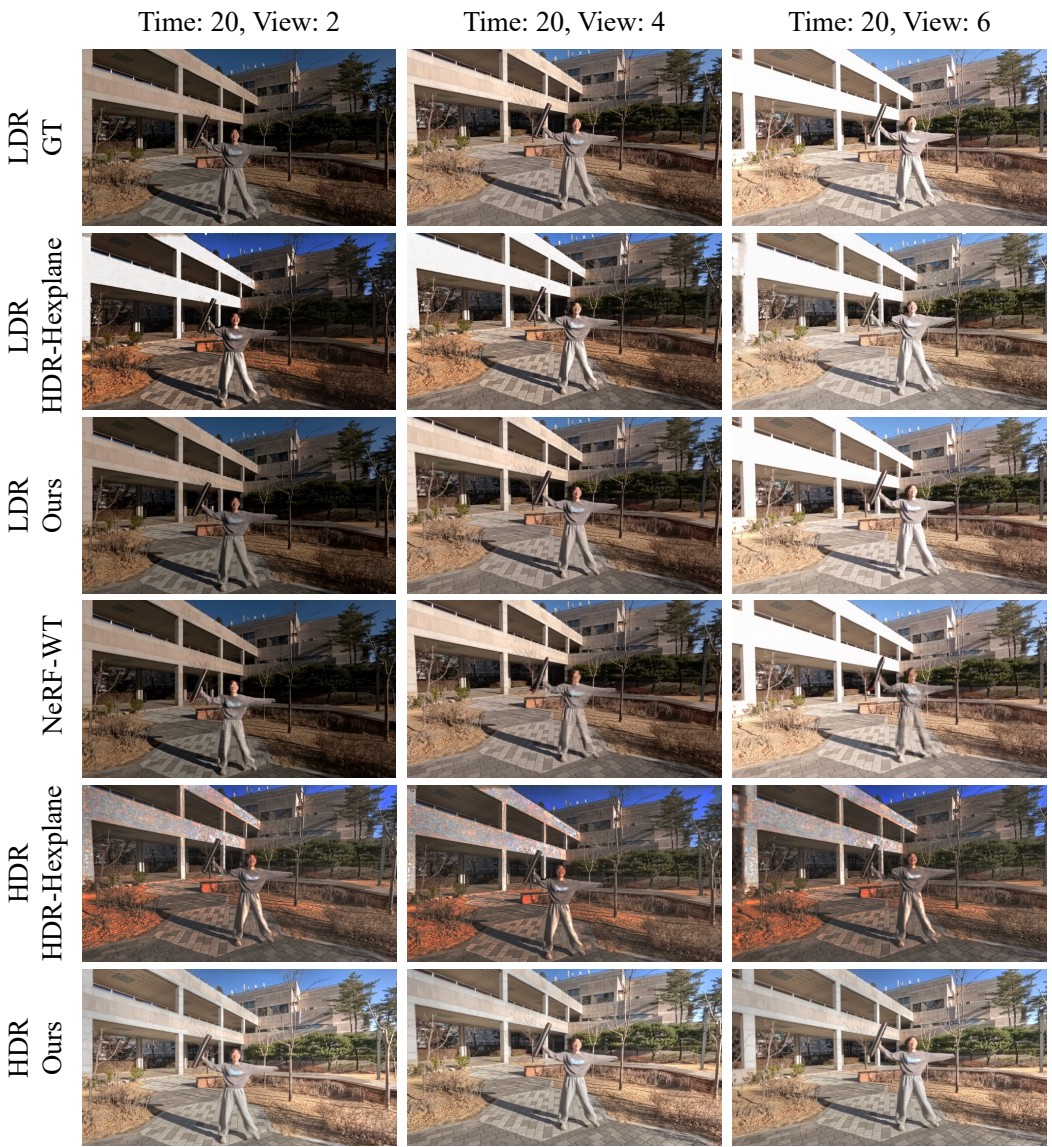

Figure S13: **Qualitative results of novel view synthesis on *Pointing walk* data.**

Time: 28, View: 1    Time: 28, View: 3    Time: 28, View: 8

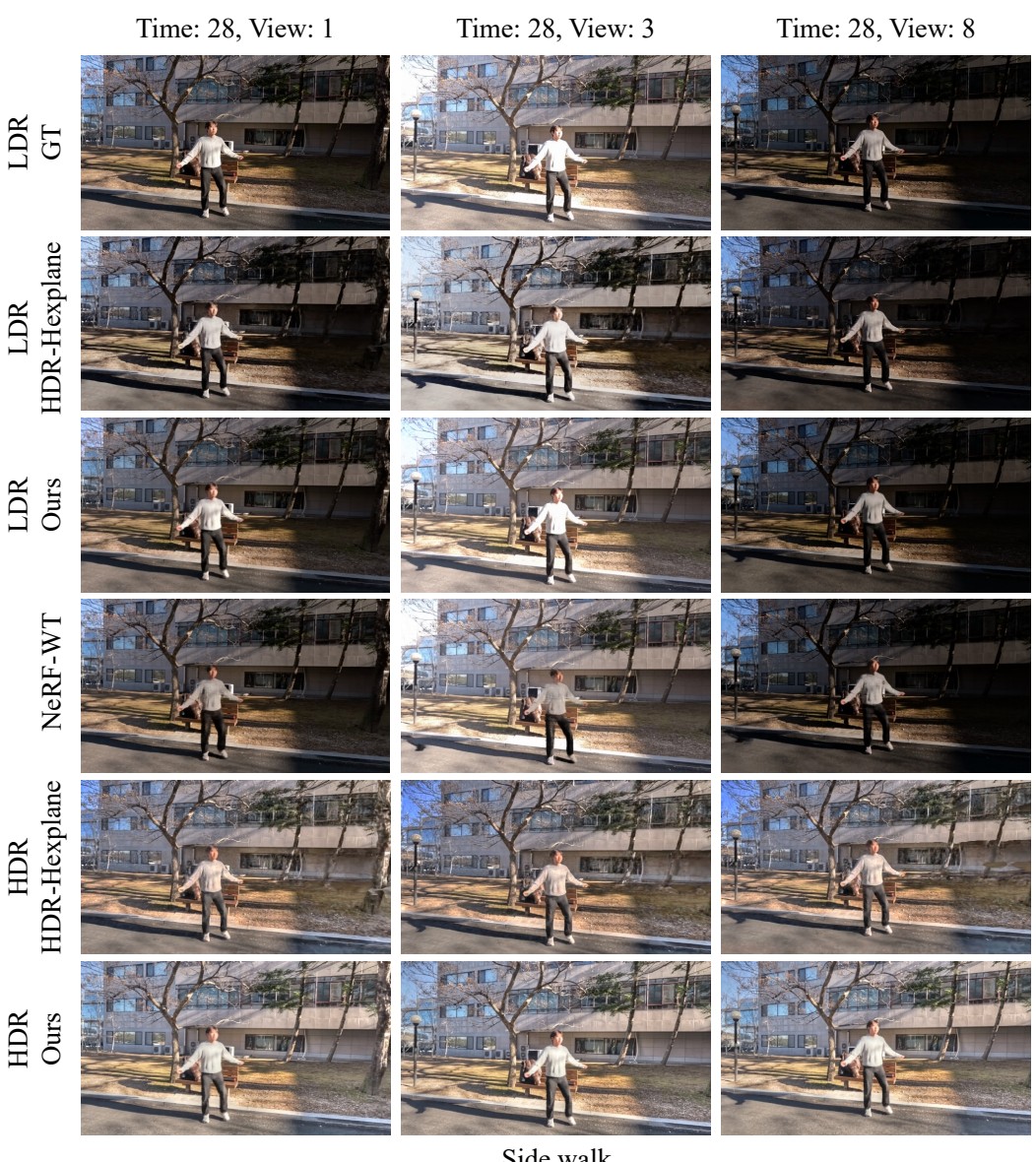

Side walk

Figure S14: **Qualitative results of novel view synthesis on *Side walk* data.**

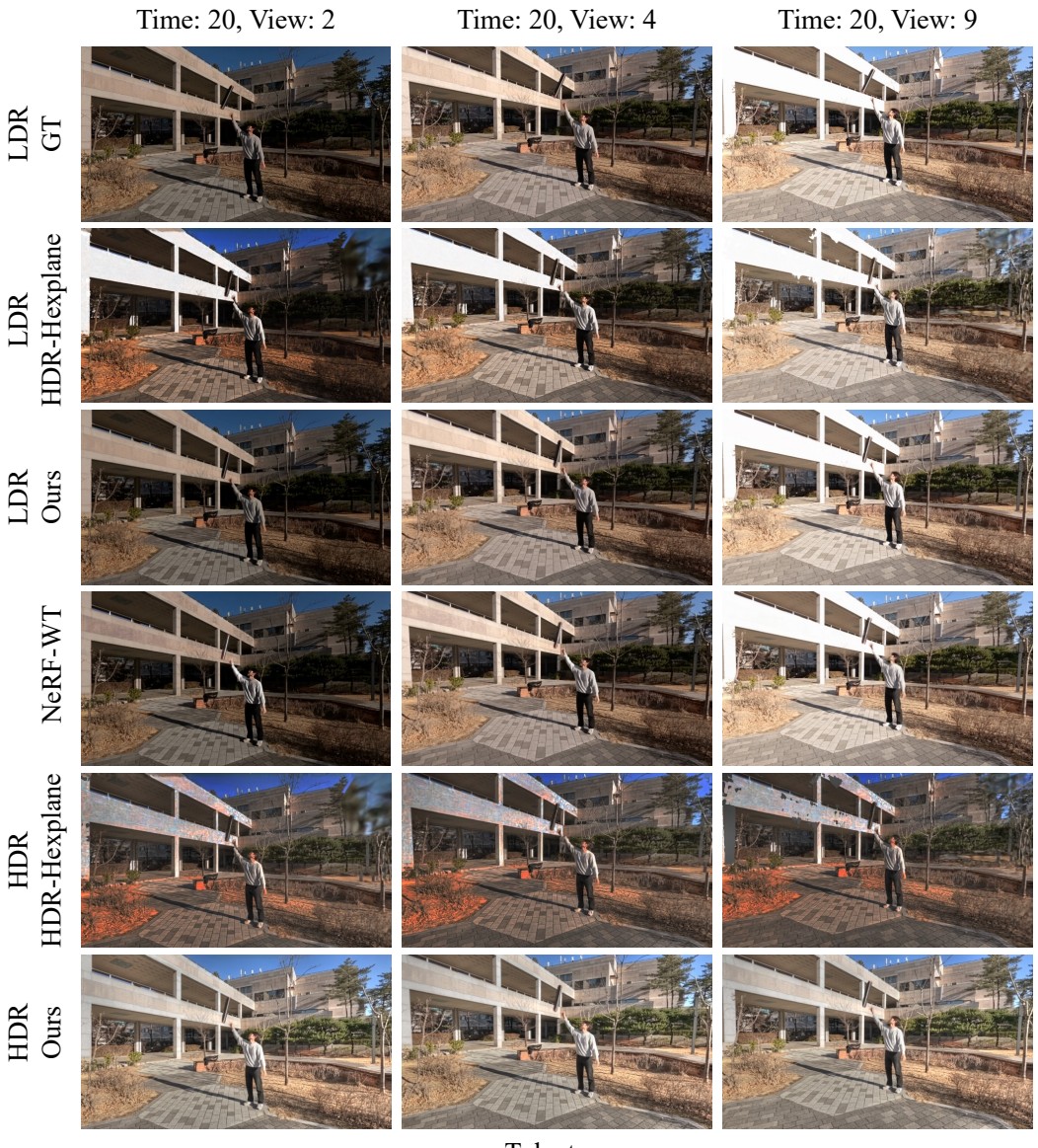

Figure S15: **Qualitative results of novel view synthesis on *Tube toss* data.**

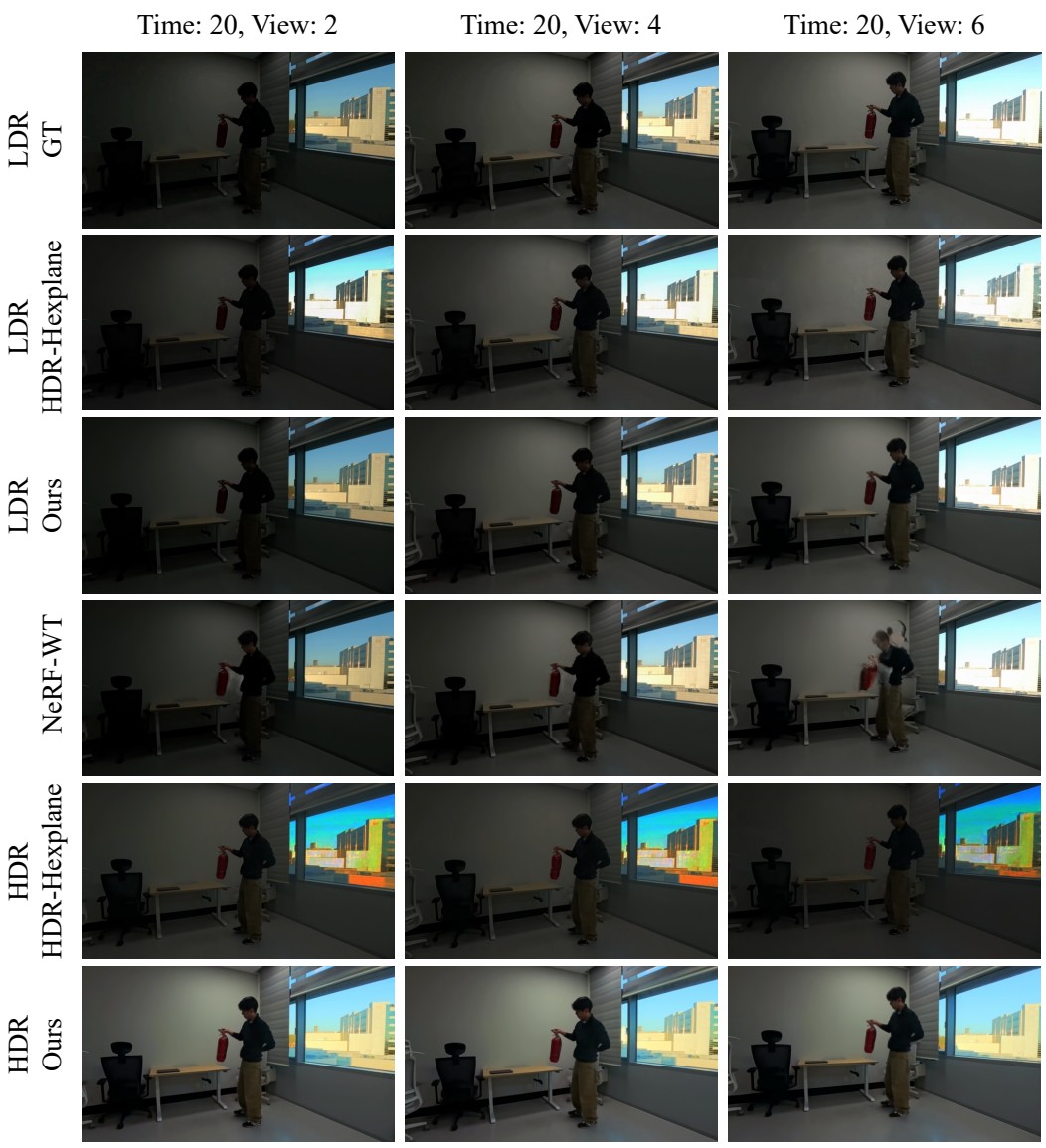

Fire extinguisher

Figure S16: **Qualitative results of novel view synthesis on *Fire extinguisher* data.**

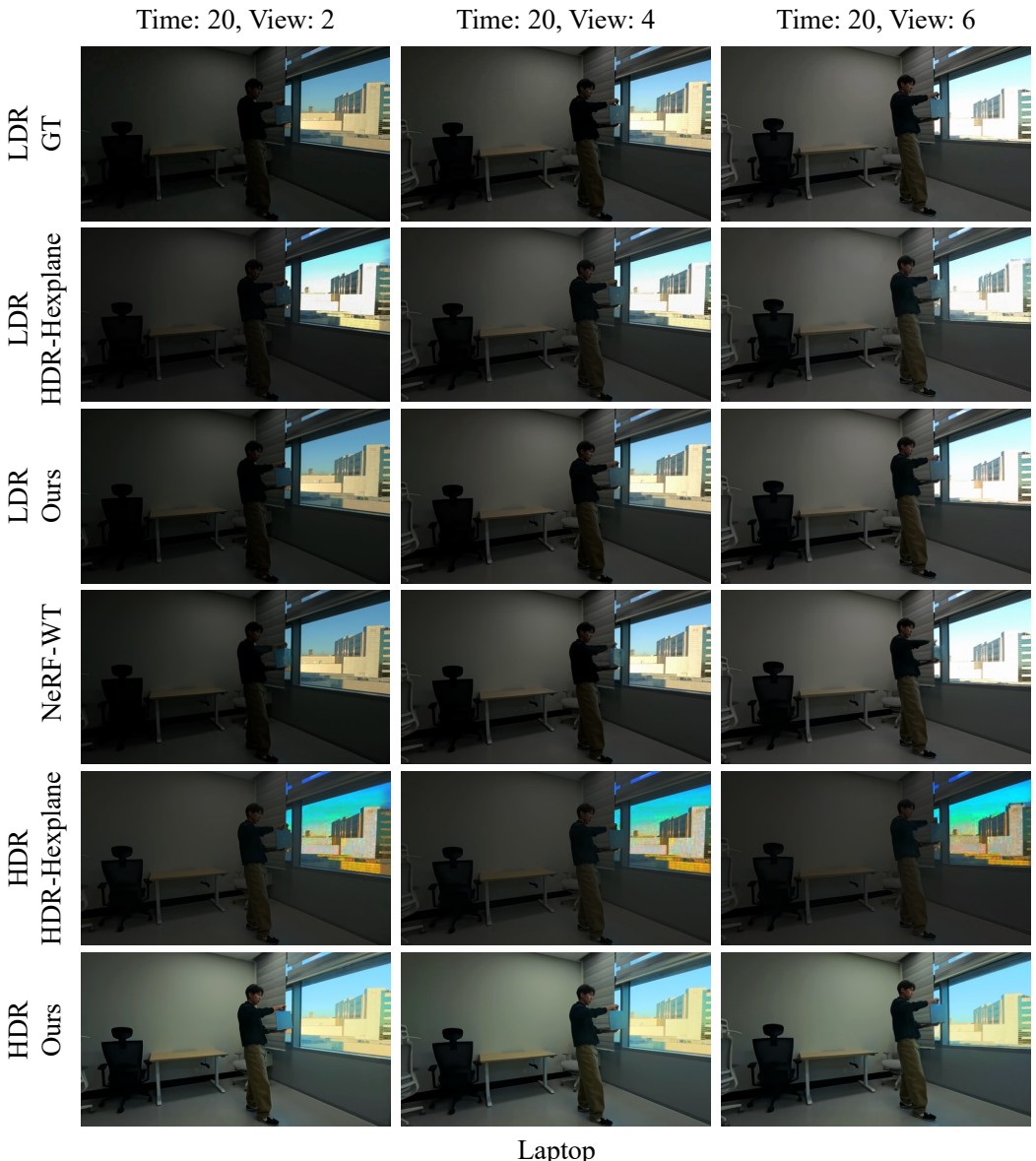

Figure S17: **Qualitative results of novel view synthesis on *Laptop* data.**

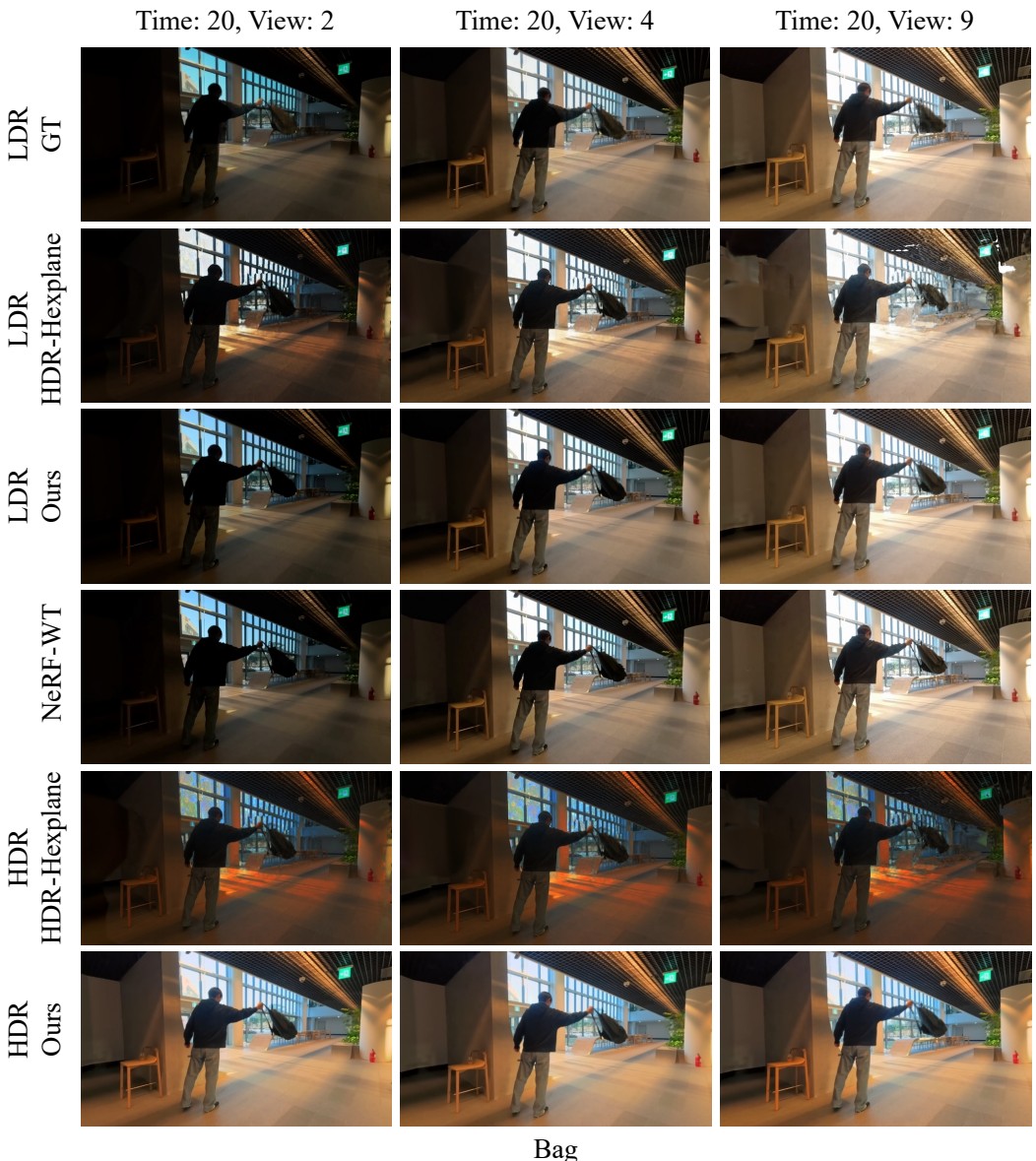

Figure S18: **Qualitative results of novel view synthesis on *Bag* data.**

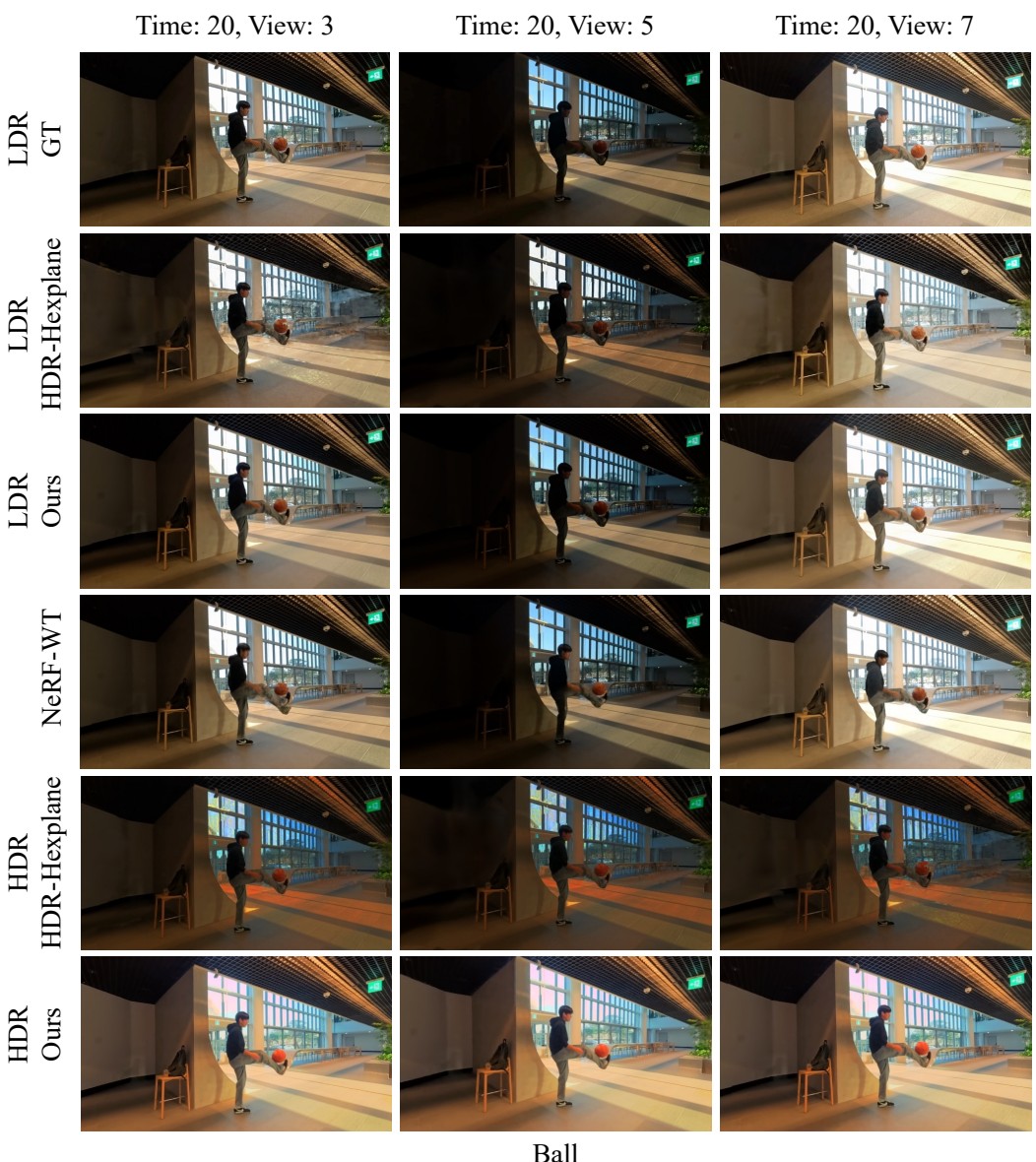

Figure S19: **Qualitative results of novel view synthesis on *Ball* data.**

## USE OF LARGE LANGUAGE MODELS

A large language model was used only for minor assistance in writing and improving the clarity of presentation.

