# OpenReview forum: "HDR-NSFF: High Dynamic Range Neural Scene Flow Fields"
_ICLR.cc/2026/Conference — ICLR 2026 Poster_

### Official Review · Reviewer_Jfnv · 2025-10-15

**Soundness:** 4
**Presentation:** 3
**Contribution:** 2
**Rating:** 4
**Confidence:** 4

**Summary:**

Paper proposes HDR-NSFF for reconstructing dynamic High Dynamic Range (HDR) radiance fields from alternatively exposed monocular videos. ​ Unlike traditional HDR methods operating on 2D images, HDR-NSFF models 3D scene flow, HDR radiance, and tone mapping in a unified end-to-end pipeline, which incorporates exposure-robust semantic flow estimation using DINO features, robust depth estimation, and generative priors to address challenges like color inconsistencies and information loss due to saturation. ​ The method is valuated on real-world and synthetic datasets, showing its outperformance over state-of-the-art methods in novel view synthesis, time interpolation, and combined view-time synthesis, achieving superior reconstruction quality and temporal coherence. ​

**Strengths:**

1. HDR-NSFF explicitly models 3D scene flow, enabling robust handling of occlusions, large motions, and dynamic changes in geometry and radiance. ​The use of DINOv2-based semantic features for optical flow estimation enhances robustness to exposure variations, addressing a key challenge in HDR reconstruction under large motion. ​

2. The proposed generative priors compensate for information loss due to saturation and sparse-view limitations. ​

3. A real-world GoPro dataset is introduced with synchronized multi-exposure captures. ​

**Weaknesses:**

1. The problem setting, capture (alternative exposures), and employment of generative methods for HDR were explored previously or similarly or at least a straightforward extension in hindsight to the current dynamic HDR for neural radiance fields.

2. The GoPro dataset may not adequately capture the diversity of real-world lighting conditions, object types, and extreme motion scenarios, as the examples demonstrated in this paper involves simple (but large motions) and often times sunny outdoor scenes, where HDR may not be  ecessarynto reveal lost details due to saturation and under exposure with noise corruption.

3. Generative priors may introduce artifacts or hallucinations, potentially affecting the fidelity of the reconstruction. ​Please point out if current experiments address this issue.

4. The framework requires significant training time (10+ hours per scene). ​

**Questions:**

Note to authors: my preliminary rating would’ve been more like 5 (currently the next rating from 4 is 6) which will be finalized during/after rebuttal.

1. Have you explored strategies to optimize the training process or reduce the computational cost for real-world applications? ​

2. How do you ensure that the generative priors do not introduce significant artifacts or hallucinations during the reconstruction process? ​ Have you evaluated the impact of these priors on the overall fidelity of the HDR radiance fields? ​

3. How well does the model generalize to other real-world scenarios with more drastic lighting conditions, object types, and motion patterns? E.g. indoor rave party scene with laser and abrupt lighting changes with multiple dancers in complex motions under full and partial occlusion.

4. How do you address inaccuracies propagated by DINOv2 and Depth-AnythingV2?

5. What are the prospects for adapting HDR-NSFF for real-time applications, such as live video processing or interactive systems?

**Details Of Ethics Concerns:**

No ethics concern.

---

> ### Author Response · Authors · 2025-11-24
> **Author response (1/2)**
>
> We thank reviewer Jfnv for the constructive feedback. We appreciate the reviewer Jfnv's recognition of our explicit 3D scene-flow modeling, the robustness of DINO-based flow, the benefit of generative priors, and the contribution of our synchronized multi-exposure dataset. Below, we address the concerns regarding novelty, dataset diversity, generative-prior fidelity, and optimization efficiency.
>
> **W1. On the concern that our framework might be a straightforward combination of existing ideas**
> >We respectfully argue that our contribution is not a trivial extension, but rather lies in the unified formulation that jointly handles:
> >- HDR radiance reconstruction
> >- 3D scene flow estimation
> >- Tone-mapping for multi-exposure inputs
> >- Exposure-robust semantic feature based motion analysis
> >- Generative priors to mitigate information loss.
>  To the best of our knowledge, no prior work has analyzed or integrated these components specifically for dynamic HDR radiance field reconstruction.
> > As several impactful works have shown (e.g., HDR-NeRF[1], Shape of Motion[2]), novel insights often arise from effective combinations of ideas tailored to a new domain.
> > Thus, the novelty is not simply the use of pretrained models or generative priors, but the synergistic framework that resolves color inconsistency, saturation-induced information loss, and exposure-variant motion simultaneously in HDR dynamic 3D scene reconstruction.
>
> >[1] Huang et al., “HDR-NeRF: High Dynamic Range Neural Radiance Fields”, CVPR 2022.
>
> >[2] Want et al., “Shape of Motion:4D Reconstruction from a Single Video”, ICCV 2025.
>
> **W2. Diversity of the GoPro HDR dataset**
>
> >While several sequences are captured in sunny outdoor scenes, the GoPro dataset is not limited to simple conditions. It includes indoor and outdoor settings, occlusions, and large camera/object motion.
> The goal of this dataset is to provide a controlled yet challenging benchmark for HDR dynamic scene reconstruction under alternating exposures—a scenario not addressed by existing datasets.
> Across all samples, our method consistently outperforms competing approaches, indicating that the proposed dataset is sufficiently challenging and effective for evaluating dynamic HDR reconstruction performance.
>
> **W3-Q2. On potential hallucinations introduced by generative priors**
> >We recognize that generative priors may introduce artifacts or hallucinations.
> To mitigate this, our framework adopts two strategies (Sec. 4.2):
> >- Delayed activation. The generative prior is disabled for the first 80% of training, ensuring that geometry and radiance are learned purely from real observations before any generative signal is introduced.
> >- Low sampling probability.After activation, enhanced views are sampled with only 10% probability, so ground-truth supervision remains the dominant signal throughout training.
> These strategies ensure that the model focuses on real, consistent structures while the generative prior functions only as a mild regularizer for saturated or missing regions.
> We have updated the revision accordingly, and the reviewer Jfnv can refer to Sec. 4.2 for the clarified description.

---

> > ### Comment · Reviewer_Jfnv · 2025-11-24
> > **Revised version?**
> >
> > While I am studying the authors rebuttal, where can I find the revised manuscript mentioned in the rebuttal? The "pdf" tab seems to be the original submission without any blue text. Thanks!

---

> ### Author Response · Authors · 2025-11-24
> **Author response (2/2)**
>
> **Q1-Q5. Reducing computational cost / toward real-time applications**
> >We agree that the current NSFF-based formulation is computationally expensive.
> However, our framework is architecture-agnostic, and its components can be applied to more efficient dynamic reconstruction backbones.
>
> > To demonstrate this, we included additional experiments in the revision where NSFF is replaced with MoSca [3], a 3D Gaussian Splatting–based method (Appendix D.2).
> Applying our exposure-robust HDR strategies to MoSca reduces the training time to about less than an hour per scene on a single NVIDIA RTX 3090, while rendering operates at real-time frame rates.
>
> > These results indicate that our HDR framework is compatible with lightweight 3DGS architectures and highlight a clear path toward real-time or interactive HDR-4D systems.
>
> > [3] Lei et al., “MoSca: Dynamic Gaussian Fusion from Casual Videos via 4D Motion Scaffolds”, CVPR 2025.
>
> **Q3. Generalization to more drastic lighting and complex motions**
>
> > We acknowledge that extremely challenging scenarios—such as abrupt laser lighting, highly non-rigid multiple actors, and heavy occlusions—remain difficult.
> > These settings often violate assumptions shared by existing dynamic NeRF/GS pipelines as well.
> > Nevertheless, reviewer Jfnv suggestion is valuable, and we believe extending HDR-NSFF to such environments is an exciting direction for future work.
>
> **Q4. Addressing inaccuracies from DINOv2 and Depth-AnythingV2**
>
> > We acknowledge that DINOv2 and Depth-Anything V2 may contain inaccuracies.
> To prevent these errors from being directly propagated, we use a progressive weighting scheme: depth and flow losses have high weights early in training to provide reliable initialization, and their influence is reduced later so the model can refine or correct these estimates through geometric regularization.
> Additionally, several regularizers (Appendix B, line 876) promote smooth and physically plausible geometry and motion, further mitigating errors from the pretrained models.

---

> ### Author Response · Authors · 2025-11-24
>
> The revised version is now available.
> Thank you for checking!

---

> > ### Comment · Reviewer_Jfnv · 2025-11-25
> > **Rating unchanged... for now**
> >
> > Thank you to the authors for their hard work on the rebuttal. After checking the revised paper, rebuttal, and other reviews, reconsidering its technical contribution, problem setting, and the results; and holistically the new delta brought to HDR, I believe my original evaluation is valid, and that the paper remains somewhat between 4 and 6 and would have been rated 5 if the review system allowed me to do so.
> >
> > Thus, I would be interested in hearing from other reviewers especially MPxi, the apparent champion of the paper, on his/her thoughts on the (revised) paper and the author's rebuttal. Until then, my rating for the paper remains unchanged for now.

---

### Official Review · Reviewer_MPxi · 2025-10-29

**Soundness:** 3
**Presentation:** 3
**Contribution:** 3
**Rating:** 8
**Confidence:** 3

**Summary:**

This paper presents HDR-NSFF, a novel framework for reconstructing dynamic HDR radiance fields from alternatively exposed monocular videos. The approach integrates 3D scene flow estimation, HDR radiance reconstruction, and tone mapping into a unified end-to-end framework. The authors also introduce a real-world GoPro dataset with synchronized multi-exposure captures to support systematic evaluation.

**Strengths:**

The paper is well-structured, and the experiments are fairly comprehensive. Reconstructing dynamic HDR radiance fields is also an interesting and meaningful idea.

**Weaknesses:**

1. The visual quality of the reconstructed images is not satisfactory; noticeable blurriness can be observed in several results (e.g., Figure 1 and Figure 6). Could the authors provide an explanation for this issue?

2. Long exposure tends to introduce motion blur (e.g., in the Big Jump scene), while short exposure often leads to noise. How do the authors address this trade-off in their method?

3. When comparing with other HDR reconstruction methods, was the same tone-mapping function used for visualization and evaluation?

4. How are the flow and depth representations integrated into the proposed system?

**Questions:**

1. Are the nine cameras perfectly parallel to each other, or are they converging toward a common focal point?

2. Were all cameras fixed and synchronized during the capturing process?

3. The proposed framework involves multiple loss terms. Is the optimization process stable, and are the chosen hyperparameters suitable for all scenes?

4. Since the method adopts a per-scene optimization strategy to reconstruct dynamic HDR radiance fields, can it generalize to render arbitrary viewpoints and timestamps?

---

> ### Author Response · Authors · 2025-11-24
> **Author response**
>
> We sincerely thank reviewer MPxi for the positive evaluation and constructive feedback. Below, we addressed the concerns raised by reviewer MPxi.
>
> **W1. On the observed blurriness in reconstructed images**
>
> >The observed blurriness mainly arises from two inherent challenges.
> First, implicit neural network such as NSFF which is NeRF-based methods tend to underrepresent high-frequency details due to the well-known spectral bias[1], which favors smooth, low-frequency solutions.
> Second, our setting—monocular, dynamic scenes with alternating exposures—is extremely ill-posed, providing sparse and inconsistent observations that further constrain the recovery of sharp textures.
>
> >Despite this, the implicit formulation offers strong temporal and exposure consistency, which is crucial for stable HDR reconstruction under varying exposures.
> For perceptual sharpness, explicit representations such as 3D Gaussian Splatting are more advantageous. To verify this, we include additional results using a 3DGS-based framework (MoSca[2]), which indeed produces sharper appearance (see `Appendix D.2`).
>
> >[1] Rahaman et al., “On the Spectral Bias of Neural Networks”, PMLR 2019.
>
> >[2] Lei et al., “MoSca: Dynamic Gaussian Fusion from Casual Videos via 4D Motion Scaffolds”, CVPR 2025.
>
>
> **W2. On handling the trade-off between motion blur (long exposure) and noise (short exposure)**
> >We appreciate this insightful question.
> >Our current method does not explicitly model motion blur or noise.
> >However, we observe that for non-extreme cases, the exposure-robust training strategies and radiance field optimization naturally mitigate much of the degradation.
> >We agree that joint HDR deburring/denoising is an interesting research direction.
> >We have added this to our discussion of future work and refer to the relevant literature:
>
> >[2] Casual3DHDR: High Dynamic Range 3D Gaussian Splatting from Casually Captured Videos. ACMMM 2025.
>
> >Thank you for the helpful suggestion.
>
> **W3. On tone-mapping consistency across comparisons**
> >For visualization, we applied the same tone-mapping operator using Photomatix Pro across all methods to allow a fair comparison.
> For novel view synthesis (NVS) evaluation, however, the target views are tone-mapped using each model’s own learned tone-mapping module. This is essential because different methods reconstruct HDR radiance fields with varying scales and distributions, and applying a single tone-mapping function would introduce mismatches.
> Thus, NVS evaluation is performed with model-specific tone-mapping to ensure a fair and faithful comparison. We have clarified this procedure in `Sec. 5`.
>
> **W4. Integration of flow and depth representations in HDR-NSFF**
> > Thank you for pointing out the need for a clearer explanation of how flow and depth are integrated.
> > The explanation is (`Appendix B`, lines 840-863) for clarity.
> > To summarize:
> >- Flow integration.
> > Scene flow is predicted in 3D and then projected into the image plane to produce 2D motion vectors (`Eq. 19`).
> > These 2D correspondences are supervised using an L1 loss against semantic flow (`Eq. 20`), enabling stable temporal alignment under alternating exposures.
> >- Depth integration.
> >Rendered depth is supervised with an L1 loss against monocular depth (`Eq. 21`), after applying scale–shift normalization to account for the inherent scale ambiguity.
> >This encourages geometry consistent with a single-view depth prior while remaining robust to exposure variations.
> Please refer to Appendix B for the detailed formulation.
>
> **Q1. Are the nine cameras perfectly parallel to each other, or are they converging toward a common focal point?**
> > The nine cameras are not converging toward a common focal point. They were manually aligned to face the same direction, forming an approximately parallel camera setup.
>
> **Q2. Were the cameras fixed and synchronized?**
> >Yes. All cameras were fixed during the capture, and temporal synchronization was achieved using the built-in synchronization software provided with the camera system.
>
> **Q3. Is the optimization stable across scenes?**
> >The hyperparameters tuned on the Big Jump sequence were used consistently for all other sequences and resulted in stable training.
>
> **Q4. Can the method render arbitrary viewpoints and timestamps?**
> > Yes. this is one of the key contributions of our work.
>  As shown in `Figure 6`, our method supports rendering from arbitrary novel viewpoints, and `Figure 7` further demonstrates rendering at arbitrary viewpoints and arbitrary timestamps.
>  Our framework enables users to revisit a captured moment from any angle and any moment in time with high-quality HDR appearance.

---

### Official Review · Reviewer_nGFz · 2025-10-30

**Soundness:** 2
**Presentation:** 2
**Contribution:** 2
**Rating:** 4
**Confidence:** 5

**Summary:**

This paper presents HDR-NSFF, a method that builds a high dynamic range random field of dynamic scenes. It proposes a framework that incorporates a learnable tone-mapping to adapt to high dynamic range. It also collects an HDR dynamic scene dataset for the task.

**Strengths:**

1. It proposes a learnable tone-mapping function for HDR reconstruction.

2. It introduces a new HDR dynamic scene dataset.

3. Experimental results are better than other SOTA methods.

**Weaknesses:**

1. I'm wondering why this paper is conducted on NSFF methods, as there are many advanced methods in NeRFs and Gaussians in recent years. Using an outdated baseline somehow weakens the contributions of the paper.

2. The writing of the method section includes so many details on pretrained model selections, whereas the main contribution about the dataset and the learnable tone mapping is relatively introduced far behind. This may mislead the reader's judgment of an important part of the paper.

**Questions:**

See above.

---

> ### Author Response · Authors · 2025-11-24
> **Author response**
>
> We thank Reviewer nGFz for the careful assessment and constructive comments. We appreciate the reviewer’s positive remarks on our learnable tone-mapping design, the introduction of a new HDR dynamic scene dataset, and the improved performance over existing state-of-the-art methods. These encouraging points highlight the relevance of tackling HDR dynamic reconstruction under challenging exposure variations.
> Below, we address the concerns regarding the choice of NSFF as the backbone and the organization of the method section.
>
> **W1. Choice of Backbone Architecture**
> >We appreciate reviewer nGFz's concern.
> >We agree that many advanced NeRF- and Gaussian-based dynamic reconstruction methods have been proposed recently.
> >Our choice of NSFF is not due to a limitation of our method, but rather a deliberate design decision.
> >To accurately reconstruct HDR dynamic scenes, the model must explicitly modeling 3D motion, as exposure alternation magnifies inconsistencies caused by inaccurate temporal alignment.
> >Among models that provide explicit scene-flow–based motion modeling, NSFF offers a stable, well-organized, and computationally lightweight backbone, making it a suitable platform for analyzing our HDR formulation and exposure-robust training strategies.
>
> >Importantly, our method is not tied to NSFF.
> >To verify generality, we additionally applied our framework to MoSca[1], a recent 4D Gaussian Splatting–based method (Appendix D.2).
> >We kept MoSca’s core algorithm unchanged and integrated our tone-mapping module and DINO-Tracker–based semantic flow.
> >Our experiments reveal that:
> >- MoSca with tone-mapping alone performs well in static regions but fails significantly in dynamic areas, indicating that exposure-variant motion cannot be learned reliably from RGB alone.
> >- When semantic flow from DINO-Tracker is incorporated, MoSca achieves stable HDR reconstruction even under challenging exposure alternation, demonstrating that our exposure-robust strategies synergize effectively with a more recent architecture.
> These results confirm that our contributions generalize beyond NSFF and are compatible with modern 3DGS-based methods.
> >Thus, NSFF serves as a representative backbone for analysis, while the core contributions of our paper remain model-agnostic and extendable.
>
> >[1] Lei et al., “MoSca: Dynamic Gaussian Fusion from Casual Videos via 4D Motion Scaffolds”, CVPR 2025.
>
> **W2. Writing structure regarding dataset and learnable tone-mapping**
>
> >We thank reviewer nGFz for this helpful suggestion.
> >Following reviewer nGFz  feedback, we have reorganized the Method section to prioritize and clearly highlight the key technical contributions.
> >Pretrained model details have been moved to later parts of the section, and additional explanations have been added where necessary.
> >We believe that this revised structure presents the main ideas more prominently and prevents potential misinterpretation regarding the importance of each component.
> >Please refer to the marks in the updated revision PDF.

---

### Official Review · Reviewer_gRTD · 2025-10-31

**Soundness:** 2
**Presentation:** 4
**Contribution:** 3
**Rating:** 4
**Confidence:** 4

**Summary:**

This paper proposes HDR-NSFF, which reconstructs an implicit 4D scene representation from a monocular dynamic scene video with random exposures, enabling the rendering of HDR novel views or controllable LDR results. The method leverages DINO-Tracker to enhance the robustness of optical flow against illumination variations and incorporates generative priors with multi-view information to address the ill-posed nature of monocular video reconstruction. Experimental results demonstrate the effectiveness of the proposed method compared to dynamic NeRF and 4DGS approaches when handling input videos with random exposures.

**Strengths:**

- The paper focuses on novel view synthesis for dynamic scenes with randomly exposed inputs, which is a relatively unexplored area in the current novel view synthesis field and holds considerable research value.
- The pipeline of the proposed method is concise and effective, and the introduced Robust Learning Strategies effectively mitigate the sensitivity of the optical flow prior to variations in input exposure.
- Experimental results demonstrate that the proposed method can handle multi-exposure input videos more effectively than dynamic NeRF and 4DGS methods.

**Weaknesses:**

1. The proposed method is built upon the NSFF framework, which relies on COLMAP to obtain camera parameters from the input images. However, when the input video exhibits large exposure variations, the quality of the camera parameters estimated by COLMAP may degrade or even cause COLMAP to fail entirely. The paper does not provide a clear solution to this issue, which may limit the practical applicability of the proposed method.
2. The paper points out that modeling HDR in 3D scenes can effectively alleviate the exposure inconsistency problem found in 2D methods. However, in the 2D HDR domain, NECHDR[1] specifically addresses the issue of exposure inconsistency in 2D approaches. Therefore, the authors should take the NECHDR work into consideration when performing HDR rendering on training views, and demonstrate through experiments that the proposed method outperforms NECHDR in terms of exposure consistency, or, if the exposure consistency performance is comparable, that the proposed method offers advantages in other aspects such as handling occlusions.
3. The proposed method should be compared with an additional pipeline: first, applying a 2D single-image or video-based HDR algorithm to the input multi-exposure video to obtain HDR results, and then using existing dynamic NeRF or 4DGS methods to represent the 4D scene based on the generated HDR video.
4. How is the ablation study in Table 3 designed? Both Dino-Tracker and the generative prior are supposed to be the key technical contributions of this paper. Why is there an entry labeled “Ours with Dino-Tracker”—does this imply that the main Ours method does not use Dino-Tracker? The authors should clearly explain the design of the ablation study. It is also recommended to conduct ablation experiments on the full model by individually removing modules such as Dino-Tracker or the generative prior. Moreover, Figure S3 in the supplementary material shows that the improvement brought by the generative prior (GP) to the visual results appears to be minimal, as the visualizations before and after adding GP look almost identical. The authors should clarify the reason for this observation.

[1] Exposure Completing for Temporally Consistent Neural High Dynamic Range Video Rendering. ACMMM 2024.

**Questions:**

In most 2D HDR methods, the scenes being processed involve small or even fixed camera motions. How does the proposed method perform on the commonly used 2D HDR datasets under such conditions?

---

> ### Author Response · Authors · 2025-11-24
> **Author response (1/3)**
>
> We sincerely thank Reviewer gRTD for the thorough evaluation and constructive feedback. We appreciate the reviewer’s recognition of (i) the novelty and value of exploring dynamic HDR novel view synthesis under random exposure settings, (ii) the effectiveness and clarity of our pipeline design, and (iii) the demonstrated advantages of our robust learning strategies over existing dynamic NeRF and 4DGS approaches.
> Here, we address the concerns raised by Reviewer gRTD.
>
> **W1. Reliability of COLMAP under exposure variations**
>
> > We appreciate the reviewer gRTD for raising this important point.
> COLMAP relies on illumination- and exposure-robust SIFT features, and its pipeline has been the standard choice across prior HDR neural field studies and benchmarks[1,2,3] when handling multi-exposure real-world data. Our dataset curation follows this widely adopted practice as well.
>
> > Nevertheless, we agree that under extreme under/over-exposure conditions, COLMAP can degrade or even fail, which is a limitation shared across all COLMAP-based HDR reconstruction pipelines.
> > We acknowledge this as an inherent limitation and will add a dedicated discussion in the revision to make this point explicit.
> Thank you for highlighting this crucial aspect.
>
> >[1] Huang et al., “HDR-NeRF: High Dynamic Range Neural Radiance Fields”, CVPR 2022.
>
> >[2] Jun-Seong et al., “HDR-Plenoxels: Self-Calibrating High Dynamic Range Radiance Fields”, ECCV 2022.
>
> >[3] Liu et al., “GaussHDR: High Dynamic Range Gaussian Splatting via Learning Unified
> 3D and 2D Local Tone Mapping”, CVPR 2025.
>
>
> **W2. Comparison with NECHDR**
>
> >We appreciate the reviewer gRTD for pointing out the relevance of NECHDR [4].
>  >Following reviewer gRTD suggestion, we have included direct comparisons with NECHDR in the revised manuscript.
> >Our experiments show that NECHDR struggles to maintain consistent HDR reconstruction when the camera undergoes non-trivial motion (see `Fig. 2`).
> >We attribute this limitation to a fundamental characteristic of 2D HDR video methods: they are primarily trained on datasets with fixed or minimal camera motion, which induces strong biases toward small viewpoint changes. As a result, when presented with significant camera motion their alignment modules fail to produce radiometrically consistent HDR frames.
>
> >In contrast, our method explicitly reconstructs HDR in dynamic 3D space with explicit scene-flow modeling, enabling geometry- and flow-aware reasoning that stabilizes radiance across time views. We have added these comparisons and the corresponding visual evidence in the revised version (see `Fig. 2`).
>
> >[4] Exposure Completing for Temporally Consistent Neural High Dynamic Range Video Rendering. ACMMM 2024.

---

> ### Author Response · Authors · 2025-11-24
> **Author response (2/3)**
>
> **W3. On the suggested baseline: “2D HDR → Dynamic NeRF/4DGS” pipeline**
>
> >We thank the reviewer again for suggesting this meaningful comparison.
> >To establish a strong and up-to-date baseline, we combine the latest 2D HDR video reconstruction models—LAN-HDR [5], HDRFlow [6], and NECHDR [4]—with MoSca [7], a recent high-performing 4D reconstruction system for monocular videos. Concretely, we first generate HDR video sequences using each 2D method and then apply MoSca for 4D reconstruction.
>
> >We perform this comparison on the GoPro dataset. The same μ-law tone-mapping operator is applied to all 2D HDR outputs before feeding them into MoSca for fair evaluation. Qualitative results compared to our model are shown in `Figure 10` of the revised manuscript.
>
> >**Note that**, since our method is built upon NSFF, we also construct an additional variant—HDR-MoSca, where our HDR-NSFF formulation is applied using the MoSca representation—for ablation and fair comparison. This extended analysis is provided in `Figure S7` in `Appendix D.3`.
>
> >Our findings show that the HDR 2D to 4D reconstruction pipeline faces several structural limitations that make it difficult to achieve stable 4D HDR reconstruction:
> >(1) Lack of 3D geometry and motion modeling
> >- 2D HDR methods operate purely in the image domain and thus do not account for 3D camera motion or scene geometry.
> Their temporal consistency mechanisms are not designed for geometry-aware exposure harmonization.
>  As a result, the generated HDR frames often exhibit geometric inconsistencies across views, which significantly degrade the subsequent 4D reconstruction stage.
>  (This corresponds to the weakness described in W2.).
>
> > (2) Propagation of 2D HDR artifacts into 4D reconstruction
> >- 2D HDR video models may introduce artifacts such as ghosting, halos, flickering, and local brightness fluctuations.
> Since NeRF/4DGS methods perform strict photometric fitting, they tend to treat these artifacts as valid observations.
> This frequently leads to distorted geometry or unstable radiance fields during optimization.
>
> >(3) Frame-dependent exposure scale inconsistencies
> >- Many 2D HDR video models produce outputs whose global intensity scales drift across time.
> Because these models reconstruct each HDR frame from only 5–7 neighboring inputs, frames that are temporally far apart often exhibit inconsistent brightness, resulting in flickering.
> NeRF/4DGS are highly sensitive to such per-frame exposure shifts, making them difficult to correct during optimization.
> In contrast, our method jointly integrates all frames into a unified 4D HDR radiance field, enabling consistent enforcement of global geometric and radiometric coherence across both views and time.
> For these reasons, the suggested pipeline struggles to provide consistent 4D HDR reconstruction under dynamic scenes with significant camera motion.
> We have included the comparative results and visual evidence in the revised manuscript (`Fig. 10` and `Fig. S7`).
>
> >[5] Chung et al., “LAN-HDR: Luminance-based Alignment Network for High Dynamic Range
> Video Reconstruction”, ICCV 2023.
>
> >[6] Xu et al., “HDRFlow: Real-Time HDR Video Reconstruction with Large Motions”, CVPR 2024
>
> >[7] Lei et al., “MoSca: Dynamic Gaussian Fusion from Casual Videos via 4D Motion Scaffolds”, CVPR 2025.

---

> ### Author Response · Authors · 2025-11-24
> **Author response (3/3)**
>
> **W4. Clarification**
>
> >(1) We have revised Table 3 to make the ablation design clearer.
> >In our updated terminology, Ours denotes the full model containing all proposed components: Dino-Tracker (DT), the generative prior (GP), and the HDR-NSFF architecture.
> >To isolate the contribution of each module, we include the following variants:
> >- Ours (w/o GP): full model without the generative prior.
> >- Ours (w/o GP + DT): model without both the generative prior and Dino-Tracker.
> This structure directly reveals the effect of each proposed component.
> > DINO-Tracker provides exposure-robust motion supervision and leads to consistent improvements in PSNR and SSIM, reflecting enhanced geometric stability under alternating exposures.
> In contrast, the generative prior primarily boosts perceptual sharpness and suppresses motion-induced blurring, which is aligned with the LPIPS improvement reported in `Table 3`.
>
> > (2) Regarding previous version of `Figure S3`, we agree that the improvement from GP appears subtle in that particular visualization.
> >This is because the viewpoint change in `Figure S3` is relatively small, where GP’s benefit is less pronounced.
> >We have replaced this example with a more representative case (now shown in `Figure 4`) where GP offers clearer advantages, particularly in large novel-view synthesis, producing sharper and more consistent HDR appearance.
> >We have incorporated these clarifications and updated visual comparisons in the revision.
>
> **Q. “In most 2D HDR methods, the scenes involve small or fixed camera motions. How does your method perform on commonly used 2D HDR datasets under such conditions?”**
>
> >This scenario indeed highlights a fundamental limitation of 3D/4D reconstruction–based approaches.
> >Most 3D reconstruction pipelines, including ours, rely on sufficient parallax to estimate scene geometry and motion reliably. >However, commonly used 2D HDR datasets exhibit small, providing far too little geometric variation for any structure-from-motion–based method to operate effectively. As a result:
> >- Such datasets do not contain enough parallax to support geometry estimation, therefore our method—and other 3D approaches—struggle to reconstruct full 3D structure under these conditions.
> >- Even if novel view synthesis is attempted, the reconstruction would be limited to a very narrow region where minimal parallax exists.
> >- Consequently, 2D HDR datasets are inherently unsuitable for evaluating 4D reconstruction–based HDR methods, as they do not provide the motion signals these methods fundamentally require.
>  For this reason, we constructed a dataset that does contain sufficient camera motion to allow fair evaluation of geometry-aware HDR reconstruction. This dataset enables us to assess the method in scenarios where its design assumptions hold.
>
> >Finally, our core motivation is to leverage 3D geometry to overcome the inherent limitations of 2D HDR approaches—such as exposure inconsistencies, temporal instability, and occlusion ambiguity—and thereby produce more consistent and physically grounded HDR dynamic scene reconstructions.

---

### Author Response · Authors · 2025-11-24

**We thank all the reviewers for their constructive comments and suggestions for improving our work. We appreciate the positive feedback:**
- Novel and valuable problem setting on dynamic HDR novel view synthesis under alternatively exposed inputs (Reviewer gRTD)
- Concise and effective pipeline with robust learning strategies mitigating exposure-sensitive flow estimation (Reviewer gRTD)
- Introduction of a learnable tone-mapping function for HDR reconstruction (Reviewer nGFz)
- Providing a new HDR dynamic scene dataset with synchronized multi-exposure captures (Reviewers nGFz and Jfnv)
- Well-structured paper and comprehensive experimental evaluation (Reviewer MPxi)
- Explicit 3D scene-flow modeling for handling occlusions and large motions (Reviewer Jfnv)
- Robust exposure-invariant flow estimation using DINO-based semantic features (Reviewer Jfnv)
- Generative priors that help compensate for saturation-induced information loss (Reviewer Jfnv)

**We thank all reviewers for their constructive feedback. We have addressed all comments in detail, and the revised manuscript includes the following major updates (revised content is marked in blue):**

- We added direct comparisons with NECHDR and incorporated the reviewer-suggested baseline that first applies state-of-the-art 2D HDR video reconstruction (LAN-HDR, HDRFlow, NECHDR) and then performs 4D reconstruction.
- We reorganized the Method section to highlight our main contributions more clearly, following the reviewers’ suggestions.
- We updated the generative-prior experiments with more representative cases where its benefits are clearly visible (Figure 4).
- We demonstrated that our exposure-robust HDR formulation is model-agnostic by integrating it into the more recent 4D Gaussian framework, MoSca, showing consistent improvements and significantly reduced training time (Appendix D.2).

**We would like to remind our key contributions:**
- We propose the first method that jointly models HDR scene flow fields enabling both novel view rendering and time interpolation.
- We enhance scene flow learning by extending DINO-Tracker for exposure-robust motion estimation, and introduce generative priors as regularizers to overcome sparse-view limitations.
- We provide extensive experiments and a new real-world dataset with alternative exposures, demonstrating state-of-the-art performance in challenging HDR scenarios.

Given these solid contributions, we emphasize that, in line with the reviewers’ assessments, our work represents a meaningful step forward for this research area, even though there is still room for further refinement. We appreciate all the insightful comments, and we believe that this revision has significantly strengthened the submission. Thank you.

---

### Author Response · Authors · 2025-12-03
**Summary of Rebuttal**

Dear AC,

We sincerely appreciate your time and effort in overseeing the review process. We also thank all reviewers for their constructive and detailed feedback. Below, we highlight the key strengths acknowledged by the reviewers and the major updates incorporated in the revised manuscript.

**Strengths highlighted by reviewers**
- Novel and valuable exploration of dynamic HDR reconstruction under randomly exposed inputs **(Reviewer gRTD)**.
- Concise pipeline with exposure-robust learning strategies **(Reviewer gRTD)**.
- Learnable tone-mapping design tailored for HDR radiance reconstruction **(Reviewer nGFz)**.
- New real-world multi-exposure HDR dynamic dataset **(Reviewers nGFz, Jfnv)**.
- Explicit 3D scene-flow modeling for handling occlusions and large motions **(Reviewer Jfnv)**.
- Strong experiments and comprehensive evaluation **(Reviewer MPxi)**.

Our revisions further reinforce these strengths through broader datasets, stronger baselines, clearer ablations, more representative visualizations, and a model-agnostic demonstration.

**Major updates in the revised manuscript**
- **Expanded Baseline Comparisons and New Experiments**
  1. **Comparison with NECHDR (Reviewer gRTD).** We added direct comparisons with NECHDR under dynamic motion and large exposure variations (`Fig. 2`).
  2. **Comparison with 2D HDR to 4D reconstruction (Reviewer gRTD).** We implemented a 2D HDR → 4D reconstruction pipeline using LAN-HDR, HDRFlow, and NECHDR followed by MoSca and compared with our end-to-end 4D HDR reconstruction method (`Fig. 10`).

- **Stronger Empirical Validation of Components**
  1. **Clear ablation study (Reviewer gRTD)**. We clarified the ablation structure isolating the effects of DINO-Tracker and generative priors (`Tables 3, 4`).
  2. **Generative prior validity (Reviewer gRTD)**. We updated visualizations with more representative cases demonstrating the perceptual and geometric benefits of our generative-prior regularization (`Fig. 4`).

- **Expanded Dataset Diversity (Reviewer Jfnv).**
  We captured four new indoor multi-exposure dynamic sequences and incorporated qualitative results in `Appendix D` (`Figs. S16–S19`), quantitative evaluation in `Tables 3, 4, S2`, and `S3`. Updated discussion confirming robust our performance across diverse indoor illumination.
This substantially strengthens the dataset diversity and validates the method beyond outdoor scenarios.

- **Reorganization and Improved Clarity (Reviewer nGFz).**
  We reorganized the `Method` section to present core contributions upfront while moving implementation details to later sections. This results in a clearer and more coherent presentation.

- **Model-Agnostic Generalization to 4DGS (Reviewers nGFz, MPxi, Jfnv).**
  To address concerns regarding the use of an NSFF backbone, potential blurriness of implicit models, and long training time, we additionally validated our framework on MoSca, a recent 4DGS-based dynamic reconstruction method.
Without modifying MoSca’s core algorithms, we integrated our tone-mapping and exposure-robust semantic flow modules (`Appendix D.2`). This yields:
  - Significant improvements over the original MoSca baseline in HDR reconstruction quality and temporal stability,
  - More reliable motion trajectories under exposure alternation, and
  - Substantial reduction in training time (≈1 hour per scene).

  These results demonstrate that our contributions are model-agnostic, generalize beyond NSFF, and are compatible with modern 4D dynamic reconstruction pipelines.

We believe that the revised manuscript thoroughly addresses all major concerns raised by the reviewers and significantly strengthens the technical clarity, empirical depth, and practical relevance of our work.
We greatly appreciate your time and thoughtful handling of our submission.


Best regards,

 Authors

---

### Meta-Review · Area_Chair_2zqV · 2026-01-08

**Summary:**

A key concern is that the framework looks like a straightforward combination of existing ideas (HDR radiance fields, scene flow, tone mapping, pretrained priors), raising concerns that the novelty might be incremental rather than fundamental.
Another concern is that the dataset lacks diversity as it may not include many real-world challenging cases.
Other concerns are effectively addressed in the authors' rebuttal.
Overall, although the technical novelty is not strong, the paper is technically solid, well-validated after rebuttal, and clearly useful to the community. This paper sits just above the acceptance threshold.

**Reviewer Concerns:**

Concerns that are largely addressed:

* Comparisons and baselines: Authors added direct comparisons with NECHDR and a “2D HDR → 4D reconstruction” pipeline, showing the advantages of the proposed method.
* Backbone generality: The authors show that the framework can also be applied to MoSca, demonstrating that the method is model-agnostic.
* Writing of the method section: The authors have reorganized the method section following the reviewer's suggestion.
* Generative prior hallucination: The authors have clarified and justified two ways (delayed activation, low sampling probability) to mitigate hallucinations.

Concerns that are still outstanding:

* The concern on technical novelty remains: The authors' response does not fully resolve the concern that the method technically combines many existing ones.
* Dataset diversity: Authors do not add new datasets. The concern on the lack of real-world challenging cases may still hold.

**Reviewer Scores:**

Reviewer gRTD

Original rating: 4

Likely change: 6

Reason: Most of gRTD’s concerns were directly addressed in the rebuttal.

Reviewer nGFz

Original rating: 4

Likely change: 6

Reason: The main concerns were directly addressed in the rebuttal.

Reviewer MPxi

Original rating: 8

Likely change: 8

Reason: Reviewer MPxi is already positive.

Reviewer Jfnv

Original rating: 4

Likely change: 4 or 6

Reason: The concerns about novelty and data diversity remain, while the concern on generative prior is better addressed.

---

### Decision · Program_Chairs · 2026-01-26

Accept (Poster)